# A metabolism-chromatin axis promotes differential ribosomal RNA transcription in the human malaria parasite

Justine E. Couble [1], Tiziano Vignolini[1], Grégory Doré[1], Selina Mussgnug[1], Bérangère Lombard [2], Michael Richard [2], Damarys Loew[2], Michael Büttner[3], Rafael Dueñas-Sánchez [3], Gernot Poschet [3], Jessica M. Bryant [4] & Sebastian Baumgarten [1] ✉

The transmission of the most virulent human malaria parasite, *Plasmodium falciparum*, relies on its survival in the contrasting environments of the human host and mosquito vector. One of the most fascinating adaptations to this lifestyle is the specific silencing of individual rDNA genes in the human host that are de-repressed following host-to-vector transmission. In this study, we define the epigenetic signatures of rRNA transcription and find that rDNA silencing relies on aerobic glycolysis, the sole energy-generating pathway in the human host. We show that disruption of $NAD^+$ regeneration during lactate fermentation promotes rDNA de-repression and identify the sirtuin histone deacetylase Sir2a as the mediator between fluctuating $NAD^+$ levels and a functional transcriptional outcome. Hence, rDNA activation appears to be coupled to the metabolic state of the parasite as it transitions from aerobic glycolysis to mitochondrial respiration during host-to-vector transmission.

Ribosomes are the macromolecular complexes of all living organisms that translate messenger RNA transcripts into proteins. Instead of comprising a homogeneous population that passively synthesize proteins, it is becoming increasingly clear that even within the same organism, ribosomes can feature remarkable compositional and structural heterogeneity, possibly giving rise to specialized functions[1] Different ribosome populations can exist in distinct cell types of multicellular organisms[1] and modifications of ribosomes in single-celled organisms can arise during changes of growth conditions[2] or in response to certain stressors[3].

A uniquely extreme example of such ribosome heterogeneity is found in *Plasmodium falciparum*, the unicellular parasite that causes the most severe form of human malaria. The parasite's life cycle between the human host and mosquito vector is driven by a tightly regulated gene expression program[4–7] that relies on epigenetic and post-transcriptional mechanisms[8]. One of the most striking

phenomena in gene transcription throughout its life cycle is the switch in expression amongst different rDNA loci[9]. In contrast to most described eukaryotic genomes, *P. falciparum* does not encode repeats of rDNA, but only five rDNA genes, each of which is located on a different chromosome[10,11]. These distinct rDNA genes can be broadly classified into A-types (A1 and A2), expressed as the sole rRNA type in the human host and S-types (S1, S2, and S3), which are additionally activated during development in the mosquito vector. This expression pattern makes *Plasmodium* one of the only known examples wherein the products of distinct rRNA genes are assembled into possibly divergent ribosomes within the same cell.

rRNA comprises up to 95% percent of total RNA in a cell, and the continuous biogenesis of ribosomes can take up to 80% of a cell's energy budget[12,13]. Thus, rRNA transcription is one of the most tightly regulated processes of any living cell, and many organisms have evolved regulatory mechanisms to precisely silence and activate rDNA

[1]Institut Pasteur, Université Paris Cité, INSERM U1347, G5 Parasite RNA Biology Group, F-75015 Paris, France. [2]Institut Curie, PSL Research University, CurieCoreTech Mass Spectrometry Proteomics, Paris, France. [3]Metabolomics Core Technology Platform, Centre for Organismal Studies, Heidelberg University, 69120 Heidelberg, Germany. [4]Institut Pasteur, Université Paris Cité, INSERM U1201, CNRS EMR9195, Biology of Host-Parasite Interactions Unit, F-75015 Paris, France. ✉e-mail: sebastian.baumgarten@pasteur.fr

genes to adjust ribosome amounts in response to nutrient availability and environmental condition[14]. In *P*. falciparum, multiple and seemingly unrelated conditions have been observed to activate silent rDNA genes in vitro, including exposure to suboptimal growth temperature (< 37 °C)[15–18], glucose starvation[15], and disruption of histone deacetylation[17]. Yet how *P. falciparum* is able to silence mosquito-stage rDNA during development in the human host and precisely de-repress them during developmental progression through the lifecycle remains unknown.

In this study, we characterize the epigenetic signature of rRNA transcription and combine assays on different parasite growth conditions with functional forward genetics to show that rDNA silencing in the human host is sensitive to NAD+/NAM ratios that are maintained through the activity of lactate dehydrogenase. In combination with quantitative chromatin immunoprecipitation and protein-DNA proximity labeling, we reveal that the NAD+-dependent histone deacetylase Sir2a promotes rDNA repression, acting as a mediator of the cell's metabolic state and local rDNA chromatin environment. With the metabolic dependence of rDNA silencing, these data provide a model of how rDNA de-repression tightly follows the natural transition from aerobic glycolysis to oxidative phosphorylation as the parasite progresses from the human host to the mosquito vector.

## Results

### rDNA transcription across the *P. falciparum* life cycle

To accurately measure the abundance of each rRNA type, we first performed total RNA sequencing across the parasite life cycle. To be able to stringently map short reads to the near-identical 28S sequences of A1/2 and S2/3, we first generated a reference genome in which the entire A1 and S3 rDNA loci are masked (see Material and Methods)[19]. Thus, when mapping sequencing reads, A2 and S2 will be representative of the A1/2 and S2/3 rDNA loci, respectively.

We find that A-type rRNA is the exclusive rRNA type throughout the development in human blood, including the asexual replicative cycle and gametocytes (Fig. 1a). Notably, the parasite expresses a mixture of A- and S-type rRNAs in the mosquito stages, suggesting that silenced S-type rRNA are de-repressed in the vector. Additionally, we identified two novel rDNA loci constituting a third rRNA class that is expressed specifically during the oocyst stage, termed 'O-type'. This type represents a minor fraction of expressed rRNA (Fig. 1b), are highly divergent from each other (Supplementary Data 1), and both surprisingly lack the 18S rDNA at their genomic loci. They are transcribed from two rDNA loci – O1 and O2 – located on chromosome 8, which brings the total number of rDNA loci in the genome to seven, all of which are located in subtelomeric regions (Fig. 1c, Supplementary Fig. 1a, Supplementary Data 1).

The putatively upstream regulatory regions of the seven rDNA loci are highly diverse in length (1.8 kb – 12 kb) and overall sequence identity (Fig. 1d, Supplementary Data 1). The upstream intergenic regions of the S2 and S3 rDNA loci are amongst the longest found in the *P. falciparum* genome and encode long non-coding (nc)RNAs[16] that share partial sequence similarity with the upstream region of the newly described O2 locus (Fig. 1d). The divergence of the rDNA upstream regions suggests that different and/or multiple regulatory mechanisms exist to regulate transcription of each rDNA locus. This is surprising given that the 18S and 28S sequences are nearly identical between A1 and A2, as well as between S2 and S3 (Supplementary Data 1). Indeed, when we compared the relative levels of rRNA transcribed from each A-type locus by counting the abundance of their defining single nucleotide variants (SNV), we found that the contribution of each A-type rDNA locus to the total pool of A-type 28S rRNA is similar between the human blood stages and oocysts, but that A1 28S rRNA levels double in the sporozoite stage (Supplementary Fig. 1b). This suggests that even near identical rDNA are subject to differential transcriptional control or post-transcriptional processing.

## Epigenetic state and chromatin organization of rDNA loci in asexual stages

A hallmark of active transcription in *P. falciparum* is histone 3, lysine 9 acetylation (H3K9ac) upstream of a transcribed gene[20]. Chromatin immunoprecipitation followed by sequencing (ChIP-seq) of H3K9ac showed a significant peak upstream of the A2 rDNA locus (Fig. 1e), but no comparable enrichment at any of the silent rDNA loci in asexually replicating cells (Fig. 1f, Supplementary Fig. 1c), identifying upstream histone acetylation as an epigenetic signature of active rRNA transcription. In parallel, re-analysis of ATAC-seq data[21] shows that the entire A2 rDNA locus is highly accessible (Fig. 1e), whereas the silent S1 and S2 rDNA loci feature only very short accessible regions upstream (Fig. 1f).

On the other hand, a conserved feature of rDNA silencing in many eukaryotes is the spread of heterochromatin along adjacent rDNA loci in repeat arrays[22]. To gain further insight into the epigenetic state of rDNA loci, we used a previously published heterochromatin protein 1 (HP1) ChIP-seq dataset[23]. Despite the lack of rDNA repeats, we find that the individual S1 and S2 silent rDNA loci, but not the active A2 rDNA locus, in asexual stage parasites are occupied by HP1 (Fig. 1e,f). Of note, HP1 occupies the S1 and S2 rDNA genic region, whereas HP1 in not detectable at the O1 locus at all and is only enriched in the upstream, ncRNA-encoding region of the O2 rDNA locus (Supplementary Fig. 1c). In addition, accessible chromatin regions and H3K9ac are not detectable at either the O1 or O2 rDNA loci (Supplementary Fig. 1c). The presence of HP1 suggests that rDNA loci must be actively de-repressed to allow rRNA transcription, and that HP1 distinguishes these loci from other silent genes that might rely only on the stage-specific presence of an activating factor (e.g. AP2s[24]). At the same time, the substantially lower HP1 enrichment at silent rDNA loci compared to adjacent (sub-)telomeric regions that are constitutively heterochromatinized (Fig. 1f, Supplementary Fig. 1d) might indicate a 'poised', facultative heterochromatic state that still enables rapid transcriptional activation.

Among all known eukaryotes, rRNA is typically transcribed by polymerase I within the nucleolus. In *P. falciparum*, rDNA loci were shown to similarly cluster in the nucleolus with DNA FISH[11]. To resolve these interactions at high resolution, we analyzed Micro-C data in blood stage asexual parasites[25]. Despite their locations on different chromosomes, we found that the two actively transcribed A-type rDNA loci in asexual stage parasites strongly interact with each other (Fig. 1g). Conversely, no direct interaction was found among any of the silent rDNA loci (Supplementary Fig. 1e). In addition, the A-type rDNA locus represents a boundary in local DNA-DNA interactions (Fig. 1h, top), indicating high levels of rRNA transcription in a compartment within the parasite's nucleus.

To identify proteins potentially involved in the structuring of the rDNA chromatin environment and/or recruitment of PolI to active rDNA loci, we annotated the PolI machinery in *P. falciparum*. This search identified sequence similarity of high-mobility group proteins 1 (HMGB1, PF3D7_1202900) and 2 (HMGB2, PF3D7_0817900) with human nucleolar transcription factor 1 (Supplementary Data 2). In humans, this protein is required for PolI recruitment and subsequent remodeling of active rDNA chromatin and rRNA transcription[26]. Moreover, related HMG-Box proteins are present in the PolI transcription initiation machinery of yeast[27,28]. We therefore epitope-tagged *Pf*HMGB1 and 2 with a C-terminal 3x hemagglutinin (3xHA) tag (Supplementary Fig. 2a–e). Immunofluorescence assay of HMGB1 showed an exclusively nuclear localization of the protein (Supplementary Fig. 2f). Subsequent ChIP-seq of HMGB1 and HMGB2 revealed that they specifically associate with the active (Fig. 1h, i, Supplementary Data 3), but not the silent rDNA loci (Supplementary Fig. 2g). These data provide evidence for functional equivalence of these proteins to other eukaryotic nucleolar transcription factors.

 2

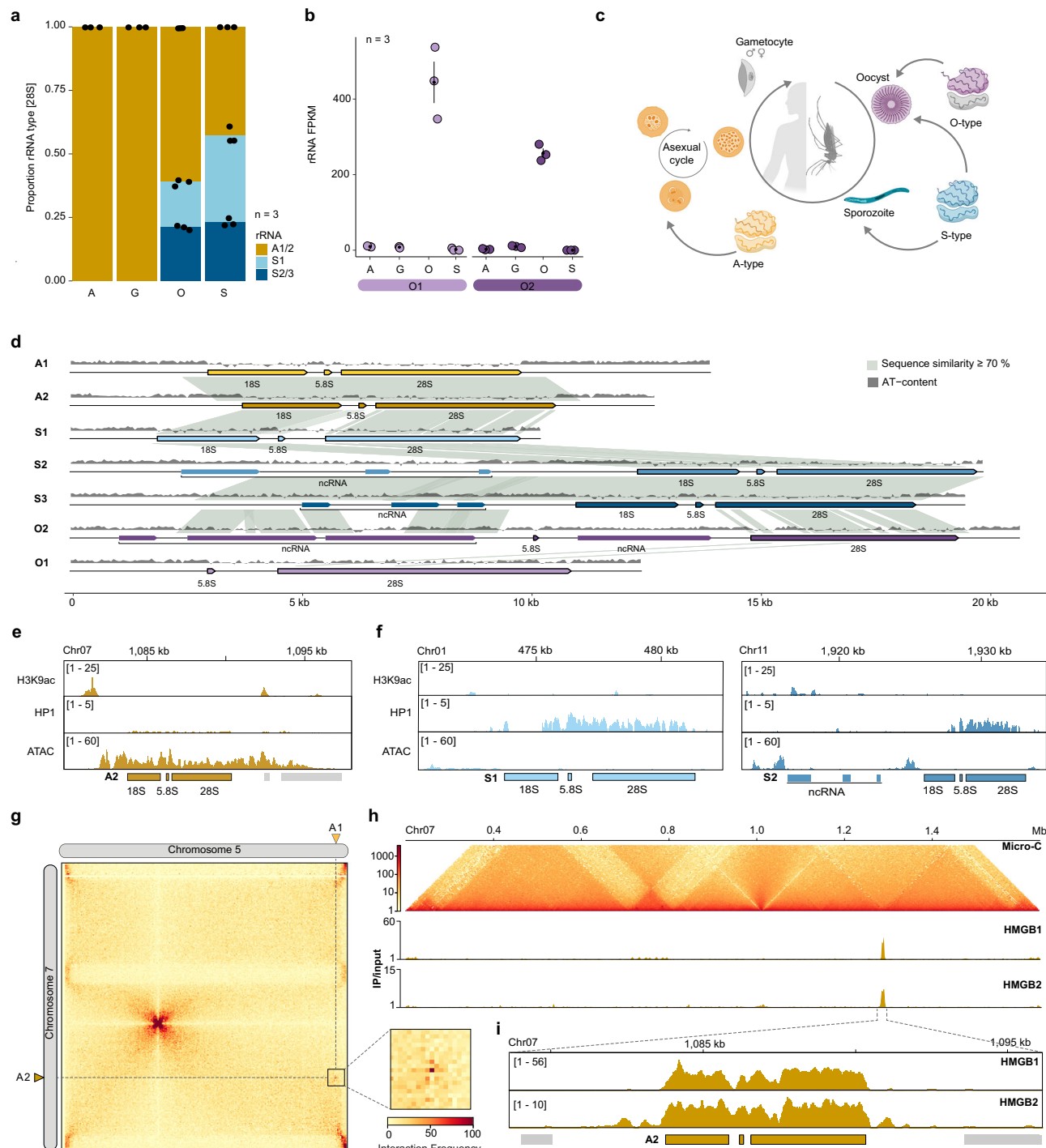

**Fig. 1 | Chromatin states of the seven *P. falciparum* rDNA loci. a** Proportion of A1/2, S1 and S2/3 28S rRNA in asexual stages (A), gametocytes (G), oocysts (O) and sporozoites (S) based on FPKM values of each rRNA type. *n* = 3 biological replicates. **b** Expression levels (FPKM: fragments per kilobase of exon per one million mapped reads) of O1 (left) and O2 (right) 28S rRNA. Black dot: mean; vertical lines: Standard error of the mean (SEM). *n* = 3 biological replicates. Life-cycle stages abbreviations as in (**a**). **c** Schematic illustration of the *P. falciparum* life cycle indicating rRNA types that are expressed in addition to the A-type rRNA at each developmental stage. Created in BioRender. Baumgarten, S. (2026) https://BioRender.com/yxykqvi.**d.** Synteny graph of the seven *P. falciparum* rDNA loci. Gray lines indicate sequence similarity ≥70% over a region of ≥100 bp. Edges of the shown regions are the start/end positions of the upstream and downstream genes. **e** H3K9ac and HP1 ChIP/input[23] together with ATAC-seq[21] ratio tracks at the active A2 rDNA locus.**f.** Same as in (**e**) for the silent S1 and S2 rDNA loci.**g.** Interchromosomal contact map of chromosome 5 and 7 at 5 kb resolution generated by Micro-C[25]. Location of A1 and A2 rDNA loci are indicated by the dotted line. Insert: Magnification of the A1/2 interaction bin (5 kb resolution).**h.** Top: Intrachromosomal contact map of chromosome 7. Middle and bottom: ChIP/input ratio track of HMGB1 and HMGB2.**i.** Detailed view of ChIP/input ratio tracks of HMGB1 (top) and HMGB2 (bottom) at the A2 rDNA locus.

### rDNA silencing is sensitive to changes of NAD$^+$/NAM levels

Decreased temperatures have been shown to promote the de-repression of silent rDNA loci in asexually replicating blood stage parasites, providing clues as to how these genes are differentially regulated between the human host and mosquito vector[15–18]. To decipher the molecular underpinnings and chromatin states of active and silent rDNA loci, we cultured asexually replicating parasites at 32 °C (rather than 37 °C). We find that at this temperature, the parasites can still complete the asexual replicative cycle (Supplementary Fig. 2h, i). Thus, this experimental system allows for precise synchronization and manipulation.

Using total RNA-seq, we first confirmed previous reports showing that S2/3-type rRNAs show the most pronounced upregulation at decreased temperatures (Fig. 2a, Supplementary Data 4). The S1 and O-type rRNA feature a similar pattern, although to a lesser degree, and A-type rRNA do not show any change in rRNA transcription (Fig. 2a). The increase in transcription of S1 and S2/3 rRNA is further accompanied by the recruitment of HMGB1 to these loci at 32 °C (Fig. 2b, Supplementary Fig. 3a, b, Supplementary Data 5), indicating that this upregulation is not a result of promiscuous transcription, but of targeted differential regulation.

To correctly detect and annotate the ncRNAs that are located upstream of the S2/3 rDNA locus and that were found to be co-regulated with the pre-rRNA at low temperatures[16], we performed full-length, native cDNA sequencing of total RNA collected at 32 °C. This technique revealed the presence of an ~8 kb transcript located upstream of the S2 rDNA locus. This ncRNA transcript is >2.5 times the size that was estimated by northern blot previously[16] (2.8 kb) and does not show any internal processing; however, it is clearly distinct from the downstream rRNA (Fig. 2b). Importantly, the ncRNA-encoding upstream region also features an increase in HMGB1 enrichment at 32 °C (Supplementary Data 5), confirming that PolI is likely responsible for its transcription[18]. A transcriptome-wide comparison of gene expression further confirmed that this ncRNA, together with the silent rRNA transcripts are among the highest upregulated genes at 32 °C (Supplementary Figs. 3c) and that there was no change in preferential transcription of the A1 versus A2 type rDNA loci (Supplementary Fig. 3d).

Given that O-type rDNA loci (1) feature a distinct chromatin landscape (Supplementary Fig. 1c), (2) have an average upregulation at 32 °C that is lower than that of S1- and S2-type 28S (Figs. 2a, and 3) each contribute at most 0.3% to the total 28S rRNA population in a cell (i.e. at the oocyst stage, Supplementary Data 4), we focused on the mechanism that promotes the de-repression of the two major silent rDNA types (i.e. S1 and S2).

To explore how a decrease in growth temperature could lead to a de-repression of silenced rDNA, we performed messenger RNA sequencing of parasites grown at 32 °C and 37 °C (Fig. 2c, Supplementary Fig. 3e, f, Supplementary Data 6). Among the most significantly downregulated genes was nicotinamidase (Nico, PF3D7_0320500), which encodes a metabolic enzyme that catabolizes nicotinamide (NAM) to nicotinic acid in the Preiss-Handler pathway[29] (Fig. 2c, Supplementary Fig. 3g). NAM itself can result from the activity of enzymes consuming NAD$^+$ as a co-substrate[30] (Supplementary Fig. 3g). We therefore compared the ratio of NAD$^+$ to NAM by targeted, quantitative metabolomics and found significantly lower NAD$^+$/NAM ratios in parasites grown at 32 °C (Fig. 2d, Supplementary Fig. 3h, Supplementary Data 7). To confirm that this is the result of decreased nico transcript levels, we directly knocked-down Nico[31] (Supplementary Fig. 3i,j). Depletion of Nico did not affect parasite growth (Supplementary Fig. 3k, i), and targeted metabolomics confirmed a significant drop in the NAD$^+$/NAM ratio between control and knock-down cells (Fig. 2e, Supplementary Fig. 4a, Supplementary Data 7). Importantly, this decrease in the NAD$^+$/NAM ratio in the Nico knock-down is accompanied by a de-repression of previously silent S-type

rRNA, with a pattern of upregulation similar to that taking place upon growth at 32 °C (Fig. 2f, Supplementary Fig. 4b, Supplementary Data 4).

While it has been known that S-type rDNA can be de-repressed by decreasing growth temperatures, our data suggest that low temperature does not directly control rRNA transcription but leads to a metabolic shift featuring decreased NAD$^+$/NAM ratios. Indeed, glucose starvation has similarly been found to de-repress silent rDNA[15], providing further evidence that it is the metabolic state that influences rRNA transcription rather than a temperature-mediated signal. On the other hand, nico transcription decreases by only half in mosquito midgut stages (Supplementary Fig. 5a), suggesting that other and/or additional factors influencing NAD$^+$ metabolism are involved in the de-repression of rRNA during the human-to-mosquito transition.

### Disruption of aerobic glycolysis de-represses silent rDNA genes

To provide further evidence for the role of metabolites – i.e. NAD$^+$/ NAM ratios - in rDNA transcriptional regulation, we investigated enzymes involved in the dramatically changing metabolism of the parasite between human blood stages and mosquito stages. During the asexual replicative cycle in the human blood, P. falciparum generates all ATP via aerobic glycolysis, or glycolysis that ends with the fermentation of glucose-derived pyruvate to lactate despite available oxygen[32] (Supplementary Fig. 3g). It follows that the single mitochondrion of the parasite is dramatically reduced, and neither the TCA cycle nor oxidative phosphorylation are essential for asexual replication in the human blood[33–37].

However, for development in the mosquito vector, the parasite extensively restructures and builds up its mitochondria[36,37]. This switch of metabolic pathways from only aerobic glycolysis to predominantly oxidative phosphorylation is accompanied by decreased expression of lactate dehydrogenase (LDH1, PF3D7_1324900), responsible for the fermentation of pyruvate to lactate, and increased expression of branched chain ketoacid dehydrogenase (BCKDHA, PF3D7_1312600), which catabolizes pyruvate to acetyl-CoA that enters the TCA cycle (Supplementary Figs. 3g, 5a). Importantly, aerobic glycolysis regenerates NAD$^+$ used in glycolysis via the activity of LDH1, maintaining elevated NAD$^+$ levels, whereas increased mitochondrial activity promotes NAD$^+$ consumption via the TCA cycle, leading to a global decrease in NAD$^+$ levels[38] (Supplementary Fig. 3g).

To test whether LDH1 is essential for maintaining cellular NAD$^+$/ NAM levels, we performed inducible knock-down of ldh1 transcripts in blood stage parasites (Supplementary Fig. 5b, c), which resulted in rapid depletion of LDH1 protein (Supplementary Fig. 5d) and an arrest in the growth cycle (Supplementary Fig. 5e). In addition, LDH1 knockdown led to a significant decrease in the NAD$^+$/NAM ratio (Fig. 2g), possibly due to the decreased rates of NAD$^+$ regeneration (Supplementary Fig. 4a, Supplementary Data 7). Moreover, we found the same pattern of silent S-type rDNA de-repression upon LDH1 knockdown (Fig. 2h, Supplementary Fig. 4b, Supplementary Data 4). Interestingly, ldh1 expression is also significantly downregulated in parasites grown at 32 °C (Supplementary Data 6), possibly leading to an additive effect together with the downregulation of nico. Altogether, this data provides further evidence that silencing of rDNA transcription in the human host is sensitive to NAD$^+$/NAM ratios and depends on the maintenance of aerobic glycolysis.

### The NAD$^+$-dependent HDAC Sir2a is required for rDNA silencing

One possible way that fluctuating NAD$^+$ levels influence transcriptional regulation of rDNA is via chromatin-modifying enzymes. NAD$^+$ is an essential co-factor in many cellular processes[39], including the activity of sirtuins, i.e. class III histone deacetylases (HDACs)[40]. The P. falciparum genome encodes two divergent sirtuin homologs, Sir2a and Sir2b. Sir2a activity depends on NAD$^+$ and at the same time is inhibited by its own product - NAM – making it sensitive to changes in cellular

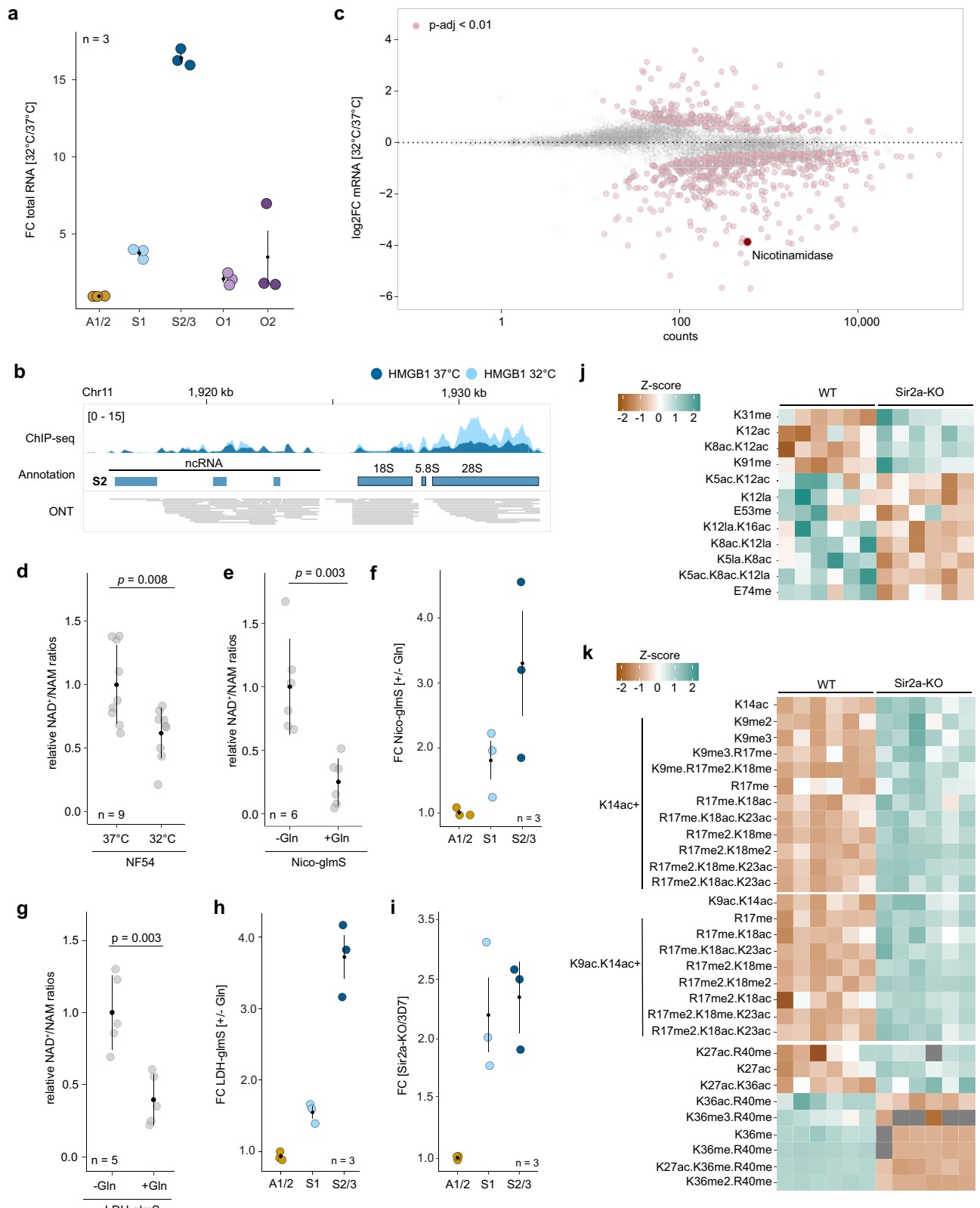

NAD+/NAM ratios[30] (Supplementary Fig. 3g). Both sirtuins were found to be involved in the regulation of heterochromatinized subtelomeric virulence genes[41–43]. In addition, Sir2a (but not Sir2b) has also been shown to repress rDNA transcription[17]. Total RNA-seq comparing wild-type[3D7] and Sir2a knock-out ('Sir2a-KO') parasites[41] (Supplementary Fig. 4b) corroborate this finding, with S-type 28S rRNA showing the strongest upregulation when Sir2a is deleted (Fig. 2i, Supplementary

Data 4). To confirm whether Sir2a could act as a 'translator' of cellular NAD+ levels on a transcriptional level, we first aimed to validate whether Sir2a can deacetylate histones in vivo. Among the histone post-translational modifications (PTMs) that were quantified using mass-spectrometry, we found that histone peptides containing H3K9ac, H3K14ac and H4K12ac were significantly and consistently more abundant in Sir2a-KO parasites compared to wild-type[3D7] parasites (Fig. 2j, k,

**Fig. 2 | rDNA repression is linked to parasite metabolism. a** Fold-change difference of rRNA expression at 32 °C compared to 37 °C for the five different rDNA loci. Black dot: mean; vertical lines: SEM. $n = 3$ biological replicates. **b** ChIP/input ratio track of HMGB1 at 32 °C (light blue) and 37 °C (dark blue) at the S2 rDNA locus. **c** Fold-change ($\log_2$, $y$-axis) between parasites grown at 32 °C compared to 37 °C plotted over the mean abundance of each gene ($x$-axis, $\log_{10}$ scale). Transcripts that are significantly up or downregulated ($p$-adj < 0.01) are displayed in pink. $n = 3$ biological replicates for each growth temperature. The FDR adjusted $p$-values were calculated in DESeq2[77]. **d** NAD⁺/NAM ratios in wild-type[NF54] parasites grown at 37 °C and 32 °C. Black dot: mean; vertical lines: SEM. $n = 9$ biological replicates. $p$-value calculated with an unpaired Welch's two-sample $t$-test. **e** NAD⁺/NAM ratios in Nico-glmS parasites grown with ( + Gln) or without (-Gln) glucosamine. Black dot: mean; vertical lines: SEM. $n = 6$ biological replicates. $p$-value calculated with an unpaired Welch's two-sample $t$-test. **f** Fold-change difference of 28S rRNA expression in Nico-glmS parasites grown with ( + Gln) or without (-Gln) glucosamine. Black dot: mean; vertical lines: SEM. $n = 3$ biological replicates. **g** NAD⁺/NAM ratios in LDH-glmS parasites grown with ( + Gln) or without (-Gln) glucosamine. Black dot: mean; vertical lines: SEM. $n = 5$ biological replicates. $p$-value calculated with an unpaired Welch's two-sample $t$-test. **h** Fold-change difference of 28S rRNA expression in LDH-glmS parasites grown with ( + Gln) or without (-Gln) glucosamine. Black dot: mean; vertical lines: SEM. $n = 3$ biological replicates. **i** Fold-change difference of 28S rRNA expression between wild-type[3D7] and Sir2a-KO *P. falciparum* parasites. Black dot: mean; vertical lines: SEM. $n = 3$ biological replicates. **j** and (**K**) Histone PTMs of H4 (**J**) and H3 (**K**) in wild-type[3D7] and Sir2a-KO parasites obtained by Mass Spectrometry. Aggregated Z-score of mean intensities of six biological replicates are represented. Only significantly up- or down-regulated peptides are shown (see Supplementary Data 8 for a complete list of PTMs). me: methylation; ac: acetylation; la: lactylation. Note: carboxyethylation and lactylation PTMs have an identical mass and cannot unequivocally be distinguished.

Supplementary Fig. 5f, Supplementary Data 8). These data provide in vivo evidence that Sir2a is a histone deacetylase.

Since no genome-wide binding data are currently available for Sir2a, we attempted ChIP-seq of an epitope-tagged Sir2a to determine whether it directly binds to rDNA loci (Supplementary Fig. 6a, b). However, recovery of Sir2a from the nuclear fraction was inefficient (Supplementary Fig. 6c). We therefore adapted a protein-DNA proximity labeling approach[44], fusing the bacterial M.EcoGII 6 mA DNA methyltransferase ('madID') to Sir2a (Supplementary Fig. 6d–f). Thus, the interaction of Sir2a with a specific genomic locus allows the madID fusion protein to add 6 mA modifications onto surrounding adenines (Fig. 3a). 6 mA sites can then be detected by direct DNA sequencing using Oxford Nanopore technology and compared to non-modified, wildtype genomic DNA. A cell line that inducibly expresses the madID protein with an N-terminal nuclear localization signal (NLS-madID) served as a negative control (Supplementary Fig. 6g–i). 6 mA could readily be detected on DNA collected from both Sir2a-madID and NLS-madID using 6 mA antibodies (Supplementary Fig. 6j). In addition, the majority of putatively modified bases identified by direct DNA sequencing in Sir2a-madID cells compared to wild-type cells were adenines (Fig. 3b). In contrast, no nucleotide preference was detected for NLS-madID, and the overall number of putatively modified sites was substantially lower, providing first evidence for increased recruitment of the madID protein in the Sir2a-madID cell line and its activity (Fig. 3b).

Interestingly, 6 mA in Sir2a-madID is not equally distributed, but is preferentially located in euchromatic regions of the genome (Fig. 3c, d, Supplementary Data 9). Whereas sirtuins in other eukaryotes are often implicated in the maintenance of heterochromatin[45], these data identify Sir2a as a possible modulator of histone acetylation across transcriptionally permissive chromatin regions. Of note, we also find specific and high Sir2a enrichment at the intron of *var* genes (Supplementary Fig. 7a), which are located in central and subtelomeric heterochromatic regions (Fig. 3c). The *var* intron represents a region known to be important for *var* gene biology, and our finding suggests that Sir2a could regulate *var* gene transcription via histone deacetylation and subsequent transcriptional repression of *var* intron-derived transcripts[46–48].

To elucidate whether Sir2a can indeed act as a modulator of histone acetylation levels at silent rDNA loci, we first performed ChIP-seq of H3K9ac, H3K14ac, H4K13ac and H4K16ac (a modification targeted by Sir2 homologs in other eukaryotes[45,49,50]) in wild-type[3D7] cells. All four histone PTMs are predominantly located within transcriptionally permissive euchromatin[20] in a pattern similar to that seen for Sir2a-madID (Fig. 3c, Supplementary Fig. 7b, Supplementary Data 9, 10). More importantly, we find that peaks of H3K14ac, H4K12ac and H4K16ac located upstream of the A2 rDNA locus are among the highest and most significant enrichments across the genome (Fig. 3e, Supplementary Data 10), similar to that seen for

H3K9ac (Fig. 1e), and that they are absent from silent rDNA loci (Supplementary Fig. 7c, d)

We next adapted a spike-in ChIP-seq approach that allows accurate quantification of changes in PTM enrichment between two samples, i.e. Sir2a-KO and wild-type[3D7] parasites[51] (Supplementary Fig. 7e). Integrating the quantitative ChIP enrichment signals with 6 mA densities showed that histone PTM peaks that feature the greatest increase of enrichment in Sir2a-KO cells compared to wild-type[3D7] cells also feature significantly higher 6 mA densities (i.e. Sir2a occupancy) than the histone PTM peaks with the lowest increase of enrichment (Fig. 3f), thus directly linking Sir2a binding with histone (de-)acetylation levels.

Most importantly, upon Sir2a deletion, we find substantial increases for all four modifications upstream of the active A2 rDNA locus, a region that also features high 6 mA densities (i.e. Sir2a occupancy, Fig. 3g, Supplementary Data 10). In direct comparison with the active rDNA locus, however, Sir2a occupancies are 2- to 13-fold higher upstream of the S-type rDNA loci (Fig. 3h, Supplementary Fig. 7f, g, Supplementary Data 11), and accordingly we find that all histone PTM levels also substantially increase in their upstream regions when Sir2a is disrupted (Fig. 3h, Supplementary Fig. 7f, Supplementary Data 11). Of note, we further find that the regions featuring the highest Sir2a occupancy generally overlap with accessible chromatin (Fig. 3h, Supplementary Fig. 7f) and—for the S2 locus—with the putative transcription start sites of the upstream ncRNA and the pre-rRNA itself (Fig. 3h). Altogether, these data indicate that high Sir2a occupancy in the upstream regions of silent rDNA loci leads to continuous histone deacetylation and maintenance of a transcriptionally repressed state.

## Discussion

The *P. falciparum* lifecycle features a highly regulated transcriptional program in which it silences specific rDNA types in the human host and de-represses them during mosquito development. The data in this study suggest that the silencing of mosquito-stage rDNA loci in the human host relies on high NAD⁺/NAM ratios that are maintained through aerobic glycolysis and translated on a molecular level by the HDAC Sir2a.

Even though the five canonical and two divergent (O-type) rDNA loci are not organized in a 'classical' rDNA repeat array and have distinct upstream regions, we identified several conserved epigenetic features of active and silent rDNA chromatin. Active rDNA loci are characterized by open chromatin, histone acetylation at regulatory upstream regions, direct interaction in 3D space, and enrichment of nucleolar transcription factors (i.e. HMGB1/2). Silent S-type rDNA loci are heterochromatinized via HP1 enrichment over their genic region and high Sir2a occupancy at their upstream regions.

Our combination of quantitative ChIP-seq of histone acetylation and Sir2a protein-DNA labeling revealed that Sir2a binds to and de-acetylates histones genome-wide, especially in euchromatic regions. Thus, histone acetylation could prove to be a key modulatory link

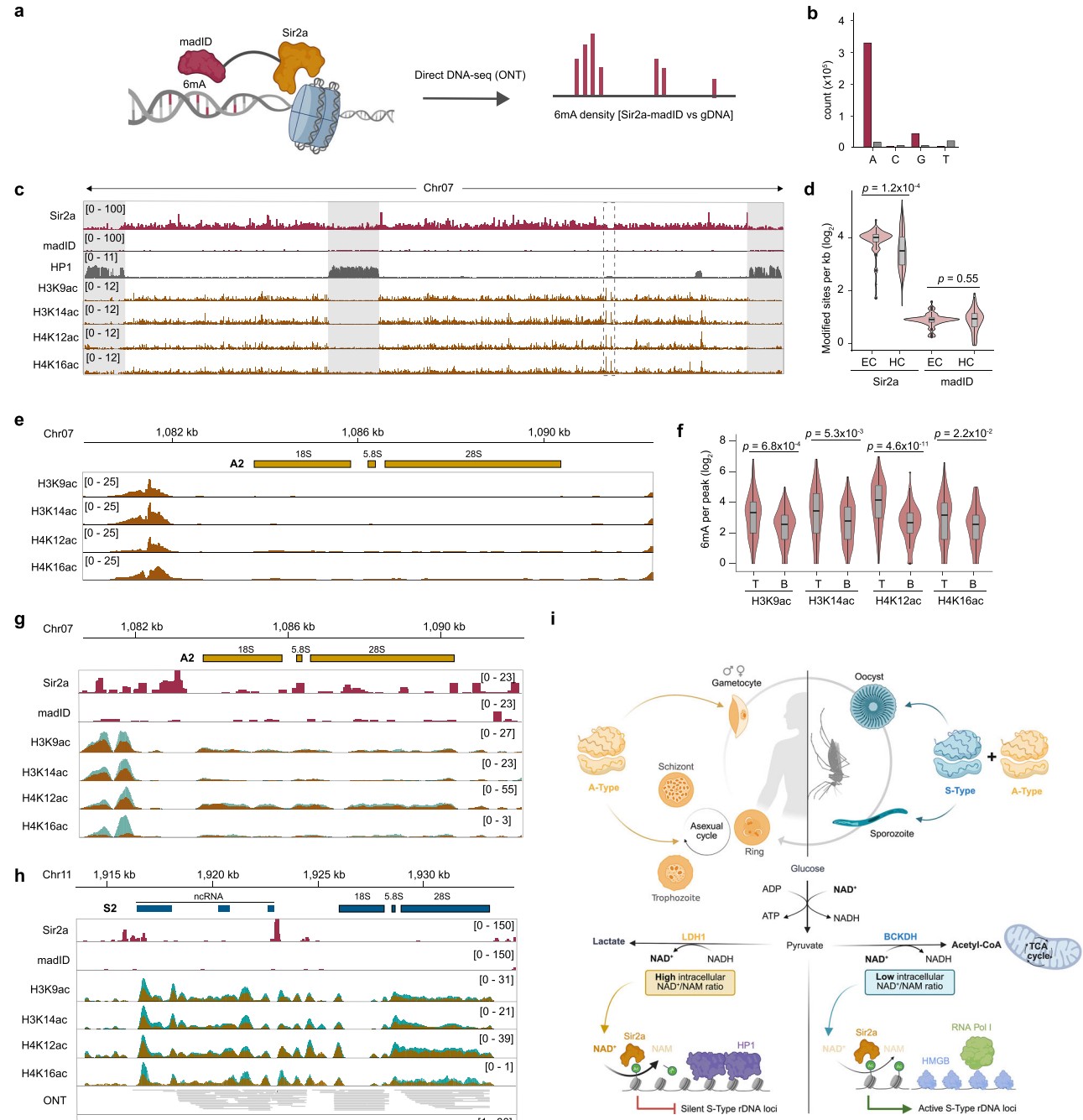

**Fig. 3 | Sir2a mediates rDNA repression. a** Illustration of the Sir2a-madID approach followed by ONT-sequencing. Created in BioRender. Baumgarten, S. (2026) https://BioRender.com/y27p272. **b** Number of modified sites in the Sir2a-madID (red) and NLS-madID (gray) cell lines categorized by nucleotide identity. A: adenine; C: cytosine; G: guanine; T: thymine. **c** Global view of 6 mA densities for Sir2-madID ('Sir2a') and NLS-madID ('madID') together with ChIP/input ratio tracks of HP1, H3K9ac, H3K14ac, H4K12ac and H4K16ac. Heterochromatic regions are highlighted in gray. Dotted insert indicates the A2 rDNA locus. madID data are shown as number of modified sites per 1 kb normalized to the AT-content. **d** Genome-wide comparison of modified sites identified in the Sir2-madID ('Sir2a') and NLS-madID ('madID)' cell lines between euchromatic regions (EC, *n* = 76) and HP1-occupied regions (HC, *n* = 89). Gray boxes represent the median (black line) and interquartile range (IQR), whiskers represent ±1.5x IQR. The *p*-value was calculated with a two-sided Wilcoxon Rank Sum test. **e** ChIP/input ratio tracks of H3K9ac, H3K14ac, H4K12ac and H4K16ac at the A2 rDNA locus. **f** Number of putatively modified sites located in the top most ('T') and least ('B') differentially

enriched peaks (*n* = 100) of each PTM that were identified by ChIP-seq in Sir2a-KO parasites (see Methods). Grey boxes represent the median (black line) and inter-quartile range (IQR), whiskers represent ±1.5x IQR. The *p*-value was calculated with a two-sided Wilcoxon Rank Sum test. **g** Detailed view of 6 mA densities in Sir2a-madID ('Sir2a') and NLS-madID ('madID') together with calibrated ChIP data for four histone PTMs. Blue: Sir2a-KO; brown: wild-type[3D7]. madID data are shown as in c). **h** Same as (**g**) for the S2 rDNA locus. Native cDNA track ('ONT') and ATAC[21] tracks are shown below.**i**. A proposed model illustrating how the changes of rRNA types are coupled to the premier metabolic pathway the parasite uses in each host. During development in the human host, NAD+ is regenerated during aerobic gly-colysis (lactate fermentation via LDH1), leading to high NAD + /NAM ratios that are used by Sir2a to deacetylate and silence S-type upstream regions. A switch to mitochondrial respiration via BCKDH lowers NAD + /NAM ratios and reduces Sir2a activity, promoting the de-repression of S-type rDNA. Created in BioRender. Baumgarten, S. (2026) https://BioRender.com/bz2mqst.

between changing environmental conditions and the transcriptional switches between rDNA loci that take place in the transition from human blood to mosquito. Sir2a might therefore not be specific for the modification of rDNA loci; however, knockout of Sir2a or the disruption of NAD+ metabolism and the subsequent increase in histone acetylation could render rDNA chromatin more permissive to binding of HMGB1, HMGB2 and subsequently PolI, which are always highly abundant and lead to a direct transcriptional response as observed in this study. This indicates that S-type rRNA transcription is not directly dependent on the developmental stage, but that the dynamics underlying their (de-)repression is linked to lifecycle progression. In contrast, other Sir2a-regulated genes might rely on the additional presence of stage-specific transcription factors (e.g. AP2s[24]) to be fully activated at different lifecycle stages.

Like previous reports[15–18], we used decreased temperature to model the upregulation of silent rRNA in asexually replicating parasites to then uncover the molecular underpinnings of rRNA regulation. In line with these earlier results, we find a consistent pattern of S-type rRNA upregulation, yet to a lower degree. This might be due to our experimental setup and model in which we used 32 °C instead of 26 °C, of which only the former allows for continuous growth and asexual replication. It is possible that lower temperatures lead to even more severe metabolic perturbations and corresponding higher rRNA de-repression, but at the cost of being able to complete the asexual replicative cycle. Although highly de-repressed at 32 °C (or 26 °C, as in previous studies[15–18]), S-type rRNA do not reach the same absolute expression levels as they normally do in mosquito stages (Supplementary Data 4). One possible explanation is the lack of mosquito stage-specific ribosome biogenesis factors and auxiliary ribosomal proteins that help process and stabilize the de-repressed S-type rRNA. In addition to the increase in enrichment of activating histone modifications and HMGB proteins, silencing histone modifications and chromatin-associated proteins (i.e. H3K9me3 and HP1) might have to be removed. Interestingly, the recently identified histone demethylases Jumonji1 and 2[52], which can remove H3K9me3, are themselves dependent on a-ketoglutarate[53]. As an intermediate of the TCA cycle, α-ketoglutarate levels might be higher in mosquito stages, allowing Jumonji demethylases to help activate H3K9me3-silenced rDNA loci. This pathway could potentially represent another link between rDNA transcription and the metabolic state of the parasite: a second regulatory layer to prevent the ubiquitously present PolI transcription machinery from transcribing silent rDNA loci during asexual replication. Thus, the metabolically-linked chromatin composition—i.e. histone acetylation versus H3K9me3—determines accessibility of rDNA loci to transcriptional activators.

While we do find that a metabolic perturbation is sufficient to render rDNA chromatin permissive to transcription, temperature itself might further represent a factor necessary for S-type rRNA homeostasis. Most importantly, we find that S- and O-type 28S rRNA sequences feature GC contents that are 11–22% lower than those of A1/2 (Supplementary Data 1). This is reminiscent of prokaryotes, where the GC content of structural RNA positively correlates with optimal growth temperatures[54]. Hence, while S-type rRNA can be upregulated at 37 °C (e.g. through LDH1 or Nico knockdowns), this temperature (or even 32 °C) might not be permissible for correct S/O-type rRNA folding and thus lower stability.

S-type transcription and possible incorporation into divergent ribosomes is a hallmark of mosquito-stage development. In return, the mechanism repressing these S-type rDNA loci is a specific mechanism of asexually replicating parasites in the human host. While our experimental model cannot recapitulate the extent of rRNA heterogeneity during mosquito development, it did allow us to define the states of active and silent rDNA loci. Furthermore, our experimental perturbations revealed the chromatin dynamics that occur when S-type rDNA loci are de-repressed and elicit a regular pattern of

quantifiable rRNA transcription. Throughout, we consistently found a specific expression pattern of the main rRNA types upon perturbation of temperature, Nico, or LDH1—an upregulation of S1 and S2/3 rRNA and no significant change in A1/2 rRNA transcription - which suggests a specific and common regulatory mechanism in all conditions and genetic backgrounds tested. We propose that the common denominator is the change in NAD⁺/NAM levels.

In this study, we combined distinct growth conditions with forward genetic approaches to independently modulate the NAD⁺/NAM ratios. Together, these data link two of the most substantial changes in the transition from the human host to the mosquito vector, i.e. the restructuring of the premier energy-generating metabolic pathway and the de-repression of silent rDNA. Key to this model is the finding that in organisms that maintain an active mitochondron but engage in aerobic glycolysis, the suppression of the latter leads to a decrease in NAD⁺ levels due to a lack of NAD⁺ regeneration by LDH1[38]. A key question for *Plasmodium* therefore is: when does the parasite naturally switch from aerobic glycolysis to oxidative phosphorylation that could lead to a decrease in NAD⁺ regeneration?

During gametocytogenesis in human blood, the parasite extensively expands its mitochondron, down-regulates LDH1, and up-regulates BCKDH. Yet inhibition of the TCA cycle does not significantly alter NAD⁺ levels at this stage, suggesting that NAD⁺ homeostasis is not primarily influenced by it[35,55,56]. Moreover, inhibition of oxidative phosphorylation does not prohibit gametocyte development[35], supporting the hypothesis that parasites enter a metabolically semi-quiescent stage during which rRNA transcription profiles do not change (Fig. 1a). Following transmission and during gamete to ookinete development (~24 h), NAD⁺ levels were found to remain stable and S-type rDNA repressed[7,57,58] (Supplementary Fig. 7h). However, genes involved in aerobic respiration and mitochondrial activity are among the most upregulated as the parasite develops into an oocyst, which coincides with the de-repression of S-type rRNA[7]. These data suggest that the parasite switches to oxidative phosphorylation at the onset of oocyst development at the same time that silent rDNA are de-repressed[7]. These data further support that temperature is not the primary factor responsible for rDNA chromatin remodeling and rRNA de-repression, given that the temperature drop already occurs much earlier, immediately after a mosquito bloodmeal.

In most eukaryotes, not all rDNA genes within an rDNA repeat are transcribed simultaneously. Because rDNA transcription, ribosome biogenesis, and protein synthesis require large amounts of energy[12,13], they need to be balanced with the resources available to the cell. As the first and rate-limiting step in this cascade, many organisms evolved mechanisms that specifically ensure the repression of rDNA transcription under unfavorable conditions (i.e. starvation, nutrient stress). This can be achieved either by targeting the PolI initiation complex[59–61] or by directly changing the chromatin environment of rDNA repeats in the nucleolus[14,62]. In the latter situation, increased NAD⁺/NADH levels during nutrient stress in human cells lead to elevated deacetylation activity of the sirtuin HDAC SirT1 (part of the eNoSC complex), resulting in heterochromatinization and silencing of rDNA[14].

Although NAD⁺ levels and sirtuin activity are central to this mechanism, a major difference with *P. falciparum* rDNA regulation is that instead of controlling the total amount of rRNA during times of nutrient stress, the parasite silences specific rDNA types as it uses distinct metabolic pathways during its progression through the lifecycle. During asexual development in the human blood, *P. falciparum* develops in a glucose-rich environment that allows rapid replication without oxidative phosphorylation, which generates more ATP per molecule of glucose than lactate fermentation during aerobic glycolysis. Yet the preference of lactate fermentation over oxidative phosphorylation is found in many cell types and organisms during periods of rapid proliferation[63] and is thought to be driven by the higher need

for $NAD^+$ than ATP, for example as a co-factor for various enzymes[38]. Given that the parasite already tightly controls the substantial restructuring from aerobic glycolysis to oxidative phosphorylation during host-vector transmission and that lactate fermentation maintains high $NAD^+$/NAM ratios, linking this switch to the (de-) repression of rDNA might therefore allow the parasite to precisely control ribosome heterogeneity without the necessity of an additional, independent regulatory mechanism (Fig. 3i). Altogether, our results provide a model of how the metabolic state of *P. falciparum* is translated into a functional transcriptional outcome to developmentally control the expression of distinct rRNA transcripts.

## Methods

### Parasite culture

Asexual blood-stage *P. falciparum* parasites were cultured in human RBCs (obtained from the Etablissement Francais du Sang with approval number HS 2021-24819) in RPMI-1640 medium (Thermo Fisher # 53400-025) supplemented with 10% v/v Albumax I (Thermo Fisher no. 11020039), hypoxanthine (0.1 mM final concentration, CC-Pro # Z-41-M) and 10 mg gentamicin (Sigma # G1397- 10 ML) at 4% haematocrit and under 5% O2, 3% CO2 at 37 °C. Parasite development was monitored by Giemsa staining. For synchronization, late-stage parasites were enriched by plasmion flotation followed by ring-stage enrichment via sorbitol (5%) lysis 6 h later. For sampling of highly synchronous parasites during the IDC, the synchronous schizonts were enriched by plasmion flotation shortly before reinvasion, followed by sorbitol treatment 6 h later. The 0 h time point was considered to be 3 h after plasmion flotation.

Gametocytes were obtained following to the protocol by Fivelman et al.[64] Briefly, synchronous asexual, late-stage parasites (30-35 h.p.i.) were concentrated at ~2.5% parasitaemia and 2.5% hematocrit using plasmion flotation. The next day, 75% of the spent culture medium was replaced with fresh medium, and the ring-stage parasites ( ~10−15% parasitemia) were left to develop into trophozoites at high parasitaemia for an additional 24 h. The culture was then diluted in fresh media to 3% parasitaemia and kept for an additional 24 h. The growth medium of the resulting high-parasitaemia, ring-stage culture was then replaced with RPMI supplemented with 5% human serum, 5% Albumax, 0.1 mM hypoxanthin, 10 mg Gentamicin and 50 mM N-acetylglucosamine (NAG, Sigma # A3286). Growth media was then changed daily for 5 days with the addition of NAG, and then without NAG for an additional 6 days. Stage V gametocytes were harvested at day 11.

### Generation of cell lines

All cell lines for genomic integration were generated with the selection-linked integration method[65] starting with the modified pSLI-sandwich plasmid as described below.

The pSLI-sandwich plasmid was digested with SalI and AvrII to remove the 2 x FKBP and GFP tags. A 3xHA tag was PCR-amplified from plasmid pUF-dCas9[66] using primers HA_F/HA_R and cloned in the digested pSLI-sandwich plasmid. To enhance the efficiency of the separation between Neomycin and the protein of interest, a GSG linker was added in between the two coding sequences. Annealed forward and reverse oligonucleotides (GSG_F/GSG_R, Supplementary Data 12) were cloned at the SalI restriction site, yielding the pSLI_3xHA_GSG_-T2A_NeoR_hDHFR plasmid. The GlmS ribozyme was amplified using primers GlmS_F/GlmS_R from the plasmid pUF-dCas9[66] and cloned into the XhoI restriction site of the plasmid pSLI_3xHA_GSG_T2A_-NeoR_hDHFR, yielding the pSLI_3xHA_GSG_T2A_NeoR_GlmS _hDHFR plasmid.

For both plasmids, the hDHFR resistance cassette was changed to a yeast dihydroorotate dehydrogenase (yDHODH) resistance cassette, conferring resistance to DSM1. To do so, the original plasmids were digested with BamHI and HindIII to excise the hDHFR cassette and the yDHODH sequence was PCR-amplified with yDHODH_F and yDHODH_R primers and cloned in the digested plasmids, yielding pSLI_3xHA_GSG_T2A _NeoR_yDHODH and pSLI_3xHA _GSG_T2A_NeoR_ GlmS _yDHODH plasmids.

The madID coding sequence[44] was codon-optimized for translation in *P. falciparum* and synthesized by GenScript. To add the fusion gene to a protein of interest, the synthesized madID gene was amplified using primers madID_F and madID_R and cloned into the AvrII restriction site of plasmid pSLI_3xHA_GSG_T2A _NeoR_yDHODH, yielding pSLI_madID-3xHA_GSG_T2A _NeoR_yDHODH.

Homology regions serving as template for the integration of the plasmid at the 3' end of the genes of interest were PCR-amplified from *P. falciparum* genomic DNA with the primer pairs HMGB1_F/HMGB1_R, HMGB2_F/HMGB2_R, Sir2a_F/Sir2a_R Nico_F/Nico_R, LDH_F/LDH_R, and Sir2a_F/Sir2a-madID_R (Supplementary Data 12). The PCR products were cloned in the NotI/AvrII digested plasmids pSLI_3x-HA_GSG_T2A _NeoR_yDHODH (for HMGB1, HMGB2, Sir2a-HA, see Supplementary Figs. 2a, 6a), pSLI_3xHA _GSG_T2A_NeoR_ GlmS_yD-HODH (for Nico and LDH1, see Supplementary Figs. 3i, 5b) and pSLI_madID-3xHA_GSG_T2A _NeoR_yDHODH (for Sir2a-madID, see Supplementary Fig. 6d)

For the NLS-madID control construct, the pFIO-hsp86_no-mCherry plasmid[67], was digested with NsiI and EcoRV. A yDHODH resistance cassette was amplified from plasmid pUF-dCas9 with primers pFIO_yDHODH_F/pFIO_yDHODH_R and cloned into the digested pFIO-hsp86_no-mCherry plasmid. The resulting plasmid was further digested with HindIII and XhoI. Annealed loxP oligos (Lox_HindIII_F/Lox_HindIII_R) and the madID-HA tag (amplified from and pSLI_madID-3xHA_GSG_T2A_NeoR_yDHODH with primers pFIO_madID_F/pFIO_ma-dID_R) were cloned into the digested plasmid in a single reaction. A NLS sequence was obtained by annealing oligos pFIO_NLS_F/pFIO_NLS_R and cloning into the HindIII restriction site of the plasmid, yielding pFIO_hsp86_NLS_madID (Supplementary Fig. 6g).

The Sir2a-KO cell line used in this study was first described in Duraisingh et al.[41]

All primers can be found in Supplementary Data 12. All PCR reactions were performed using the KAPA HiFi DNA Polymerase (Roche # 07958846001) following the manufacturer's protocol, but with lower elongation temperature (62 °C or 68 °C). Cloning and plasmid amplification were performed using the In-Fusion HD cloning kit (Clontech # 639649) and XL10-Gold ultracompetent *E. coli* (Agilent Technologies # 200315) following the manufacturer's protocol.

All plasmids were sequenced either by Sanger or Nanopore full-plasmid sequencing to verify that no mutation appeared. All pSLI plasmids were transfected into ring-stage *P. falciparum* (strain NF54) parasites by electroporation following standard protocols[66] and using 100 μg plasmid. The pFIO_hsp86_NLS_madID was transfected into the *P. falciparum* NF54^DiCre cell line[68]. To select for plasmid uptake, transfected parasites were cultured with 1.5 μM DSM1 (MR4/BEI Resources) and drug-resistant parasites emerged after three to 4 weeks. Positive selection for integration of the pSLI plasmids was performed via the addition of 400 μg/mL G418 (Sigma # G8168). For all cell lines, parasites were collected by saponin lysis (0.075% in DPBS) and gDNA was extracted using the Qiagen DNeasy Blood & Tissue Kit (Thermo Fisher # 69504). Plasmid integration was verified by PCR with corresponding primers listed in Supplementary Data 12. Since no specific primers for the verification of successful integration of the HA-tag at the HMGB1 locus were found, integration was verified by DNA sequencing using Oxford Nanopore sequencing (see below and Supplementary Fig. 2b)

### Parasite growth assay

To measure parasite growth kinetics the cell lines were tightly synchronized by plasmion/sorbitol to a 6 h window as described above. The ring-stage parasites were diluted separately to 0.2% parasitemia (5% hematocrit) in the blood of three different donors. For the Nico-

glmS cell line, the culture was split and glucosamine (Sigma # G1514) was added to one half of the culture (2.5 mM final concentration). The growth curve was performed in a 96-well plate (200 μl culture per well) with three technical replicates per condition and per blood. Parasitemia was measured every 24 h by staining parasite nuclei using SYBR Green I (Sigma # S9430) and quantifying infected RBCs using a Cytoflex flow cytometer. For the growth curve at 32 °C and 37 °C, the culture was split after synchronization, and additional timepoints were collected at the time of schizont rupture to precisely measure the duration of the life cycle at the different temperatures. All data were post-processed and analyzed using FlowJo (Supplementary Data 13, Supplementary Fig. 8).

Since knockdown of LDH-glmS following addition of glucosamine at the ring-stage did not allow for the completion of one full cell cycle, growth dynamics were compared by Giemsa-staining. The parasites were tightly synchronized by plasmion/sorbitol to a 6 h window and diluted to 1% parasitemia (5% hematocrit). The culture was then split, and glucosamine was added (2.5 mM final concentration) to one half after the sorbitol. Parasite growth was monitored by Giemsa staining every 12 h for 74 h (Supplementary Fig. 5e).

## Western Blot

For HMGB1-HA and HMGB2-HA cell lines (Supplementary Fig. 2d, e), synchronized late-stage parasites were collected by saponin lysis (0.075% saponin [Sigma # S790] in Dulbecco's phosphate-buffered saline [DPBS, Thermo Fisher # 14190-144]) at 37 °C. Parasites were washed twice with ice-cold DPBS, and lysed by resuspending the cell pellet in TLB (Total Lysis Buffer: 20 mM Tris-HCl pH 7.5, 50 mM NaCl, 1 mM DTT, 0.1% SDS and protease inhibitor [PI, Roche # 11836170001]) and sonicated with a Bioruptor Pico (5 cycles 30 s ON / 30 s OFF) at 4 °C. Cell debris was removed by centrifugation (13,500 g, 10 min, 4 °C) and the supernatant was transferred to a new tube. 25 μl of protein G Dynabeads were washed twice with TLB, resuspended in TLB with 1 μl of anti-HA antibody (Abcam # ab9110) and rotated for 2 h. The beads were washed twice with TLB, and added to the HMGB1/2 protein lysates. The HA-tagged proteins were bound by overnight rotation, the beads were washed twice with TLB and immunoprecipitated proteins were eluted by resuspending the beads in TLB supplemented with NuPage Sample Buffer (Thermo Fisher # NP0008) and NuPage Reducing Agent (Thermo Fisher # NP0004) and incubation at 70 °C for 10 min.

For the Nico-glmS cell line (Supplementary Fig. 3l), glucosamine was added to half of the culture immediately before the synchronization by plasmion/sorbitol. Cells were collected after 24 h and 48 h of glucosamine treatment and lysed with 0.075% saponin in DPBS at 37 °C. Cells were washed twice with DPBS at 4 °C. For separation of the cytoplasmic and nuclear protein fractions, the cell pellet was first resuspended in 1 ml CLB (Cytoplasmic Lysis Buffer :25 mM Tris-HCl pH 7.5, 10 mM NaCl, 1% IGEPAL CA-630, 1 mM DTT, 1.5 mM MgCl2, 1xPI) and incubated on ice for 30 min with regular flicking. The cytoplasmic lysate was cleared by centrifugation (13,500 g, 10 min, 4 °C). The nuclei pellet was gently washed twice with CLB, resuspended in 100 μl NLB (Nuclear Lysis Buffer: 25 mM Tris-HCl pH 7.5, 1 mM DTT, 1.5 mM MgCl2, 600 mM NaCl, 1% IGEPAL CA-630, PI) and sonicated with a Bioruptor Pico (5 cycles 30 s ON / 30 s OFF). This nuclear lysate was cleared by centrifugation (13,500 g, 10 min, 4 °C). Protein samples were supplemented with NuPage Sample Buffer and Reducing Agent and denatured for 10 min at 70 °C.

The LDH1-glmS cell line (Supplementary Fig. 5d), parasites were synchronized by plasmion/sorbitol, and glucosamine (2.5 mM final concentration) was added immediately after the sorbitol to half of the culture. Cells were collected after 6 h, 24 h and 48 h of glucosamine treatment, lysed and proteins were extracted with TLB as described above. Protein samples were supplemented with NuPage Sample Buffer and Reducing Agent before denaturation for 10 min at 70 °C.

For the Sir2a-HA cell line (Supplementary Fig. 6c), late-stage parasites were collected and lysed with 0.075% saponin in DPBS at 37 °C. Cells were washed twice with DPBS at 4 °C. The separation of the cytoplasmic and nuclear protein fractions was performed as described above, using CLB and NLB. Protein immunoprecipitation was performed using anti-HA antibody (Abcam # ab9110) with overnight rotation in CLB for cytoplasmic fractions and NLB for nuclear fractions. Immunoprecipitated proteins were eluted by resuspending the beads in NLB or CLB supplemented with NuPage Sample Buffer and NuPage Reducing Agent and incubation at 70 °C for 10 min.

For NLS-madID (Supplementary Fig. 6i), the cell line was synchronized by plasmion-sorbitol to a 6 h window. Rapamacin was added (200 nM final concentration) to half of the culture after the sorbitol, and cells were collected at 40 h.p.i. The separation between nuclear and cytoplasmic fractions was performed as described above.

For all Western Blots, the samples were separated on a NuPage 4–12% Bis-Tris gel (Thermo Fisher #NP0321) using MOPS running buffer (Thermo Fisher # NP0001) at 100 V for 1.5 h and transferred to a PVDF membrane using Trans-Blot Turbo transfer system (Bio-Rad) using the mixed molecular weight program. The membrane was blocked for 1 h in 1% milk in 0.1% Tween20 in PBS (PBST). Histone H3 was detected with anti-H3 (Abcam no. ab1791: 1:1,000 in 1% milk-PBST) primary antibody, followed by donkey anti-rabbit (GE # NA934-1ML) secondary antibody conjugated to HRP (1:5,000). HA-tagged proteins and PfAldolase were detected using HRP conjugated anti-HA (Ozyme 14031S, 1:1,000 in 1% milk PBST), and anti-PfAldolase (Abcam # ab38905, 1:5,000 in 1% milk PBST) antibodies, respectively. The HRP signal was developed using the SuperSignal West Pico chemiluminescent substrate (Thermo Fisher # 34580) and imaged with a ChemiDoc XRS+ (Bio-Rad).

## Immunofluorescence assay (IFA)

1 ml resuspended HMGB1-HA parasite culture was spun down and the cells were fixed in freshly prepared 4% paraformaldehyde, 0.0075% glutaraldehyde in DPBS (Thermo Fisher # 14190-144) for 30 min, rotating at room temperature. Free aldehyde groups were quenched by a 10 min-incubation in freshly prepared 50 mM NH4Cl at room temperature. The cells were washed with DPBS and stored in DPBS at 4 °C. For performing the IFA, 150 μl cells were spun down, washed once again with DPBS. Subsequently, the cells were permeabilized by an incubation in freshly prepared 0.1% TritonX-100 for 10 min rotating followed by three washes with DPBS. Afterwards, the cells were blocked for 60 min with 3% BSA in DPBS, rotating. After 60 min blocking, the cells were incubated with the HA-antibody (rat-anti-HA [clone 3F10, Sigma # 12158167001], 1:500 in 3% BSA in DPBS) with rotation for 90 min. The cells were washed 3×10 min with DPBS, rotating. Then the cells were incubated with the secondary antibody (Alexa Fluor 488 goat anti-rat IgG (H + L), Thermo Fisher # A11006, 1:2000 in 3% BSA in DPBS) and Hoechst 33342 (1:500, Thermo Fisher # H3570) and covered in aluminum foil from now on to protect from light. After the incubation, the cells were washed 3 × 10 min with DPBS, rotating. The cells were spun down and carefully resuspended in ~80 μl Vectashield Antifade Mounting Medium (Vector Laboratories # H-1900) and put on self-made Poly-D-Lysine slides. The slides were examined the day after to allow the cells to settle down over night. Confocal microscopy was performed on a Zeiss laser scanning confocal microscope (LSM810) with 63 X oil immersion objective. Acquired images were processed in Fiji.

## Histone mass-spectrometry

**Histone extraction.** *P. falciparum* (strain 3D7) and Sir2a-KO cell lines were tightly synchronized using plasmion/sorbitol and 6 replicates, each corresponding to ~2 × 10⁻⁸ parasites, were collected at 40 h.p.i. (Supplementary Fig. 5f) and lysed with 0.075% saponin in DPBS. The pellet was transferred to 1.5 ml Protein LowBind Eppendorf tube,

washed twice with 1 ml cold DPBS (4,000 g, 5 min, 4 °C), snap-frozen and stored at -80 °C until ready to be processed. The pellet was resuspended in 500 μL of cold buffer A (15 mM Tris–HCl pH 7.5, 60 mM KCl, 15 mM NaCl, 5 mM MgCl$_2$, 1 mM CaCl$_2$, 250 mM sucrose, 0.3% NP40, PI) and incubated on ice for 10-20 min, flicking the tube regularly. The pellet was then washed twice with buffer A without NP40 to remove detergent (1000 g, 5 min, 4 °C) and 5X pellet volume of 0.25 M HCl was added. The tube was rotated for 4 h at 4 °C, centrifuged (3400 g, 5 min, 4 °C) and the supernatant containing histones was transferred to a new tube. A second spin was performed (3400 g, 5 min, 4 °C) to remove pellet traces and the supernatant was again transferred to a new tube. Tricholoroacetic acid was added to a final concentration of 20% and the sample incubated on ice for 1 h. After spinning (3400 g, 5 min, 4 °C), the supernatant was carefully removed and washed with 500 μl acetone + 0.1% HCl. The histones were then washed twice using 100% acetone. The pellet was dried for 30 min under a chemical hood, and resuspended in 40 μl of loading buffer (NuPage Sample Buffer and NuPage Reducing Agent in ddH$_2$O). The samples were then loaded on a 15% home-made acrylamide gel and ran at 30 mA in SDS-PAGE running buffer. The gel was stained with LabSafe GEL Blue (VWR 786-35) for 1 h and washed with distilled water until clear. Gel slice corresponding to histones were excised and in-gel digested by using Tryps/LysC (Promega). Peptides extracted from each band were loaded onto homemade C18 StageTips (packed with AttractSPE Disk Bio C18, Affinisep) for desalting. Peptides were eluted using 40 / 60 acetonitrile / H2O + 0.1% formic acid and vacuum concentrated to dryness. Peptides were resuspended in loading buffer (0.3% TFA in miliQ water) before liquid chromatography-tandem mass spectrometry (LC-MS/MS) analysis.

**LC-MS/MS Analysis.** Online LC was performed with an RSLCnano system (Ultimate 3000, Thermo Scientific) coupled to an Orbitrap Exploris 480 mass spectrometer (Thermo Scientific). Peptides were first trapped onto a C18 column (75 μm inner diameter × 2 cm; nano-Viper Acclaim PepMap™ 100, Thermo Scientific) with buffer A (2 / 98 acetonitrile / H$_2$O + 0.1% formic acid) at a flow rate of 2.5 μL/min over 4 min to concentrate the samples. Separation was performed on a 50 cm nanoviper column (i.d.75 μm, C18, Acclaim PepMap™ RSLC, 2 μm, 100 Å, Thermo Scientific) regulated to a temperature of 50 °C with a linear gradient of 2% to 30% buffer B (100% acetonitrile, 0.1% formic acid) at a flow rate of 300 nL/min over 91 min. MS full scans were performed in the ultrahigh-field Orbitrap mass analyzer in ranges m/z 375–1500 (resolution of 120000 at m/z 200; maximum injection time 25 ms; AGC 300%). The top 20 most intense ions were subjected to Orbitrap for further fragmentation via high energy collision dissociation (HCD) activation and a resolution of 15000 with the auto gain control (AGC) target set to 100%. We selected ions with charge state from 2+ to 6+ for screening. Normalized collision energy (NCE) was set at 30 and the dynamic exclusion of 40 s.

**Histone PTM analysis.** For PTM identification, the data were searched against the *Plasmodium falciparum* histone sequences using Mascot. Only peptides that could be specifically assigned to a single histone (i.e. are proteotypic) were included in the analysis. Enzyme specificity was set to trypsin and a maximum of five-missed cleavage sites were allowed. Oxidized methionine, carbamidomethyled cysteine, N-terminal acetylation, acetylation, methylation (mono, di and tri), ubiquitination, propionylation, butyrylation, succinylation, malonylation, hydroxybutyrylation, glutarylation and crotonylation, palmitoylation, carboxyethylation of lysine, methylation (mono and di) of arginine, monomethylation of glutamic acid and aspartic acid were set as variable modifications and with a maximum of nine modifications for all Mascot searches. Maximum allowed mass deviation was set to 10 ppm for monoisotopic precursor ions and 0.02 Da for MS/MS

peaks. Importantly, lysine carboxyethylation and lactylation PTMs have an identical mass and cannot unequivocally be distinguished.

For post-translational modifications (PTMs) quantification, Skyline was used for processing the data (version 23.1.0.380, MacCoss Lab Software, Seattle, WA; https://skyline.ms/project/home/software/Skyline/begin.view) and extracted

Ion chromatograms from each peptide ion and peak area were integrated. Peptides were grouped according to their specific protein sequences and then PTMs quantifications were run independently for each peptide group (Supplementary Data 8). The resulting files were further processed using myProMS v3.10[69] (https://github.com/bioinfo-pf-curie/myproms). For each site, the peak areas of corresponding ions were log$_2$-transformed, and the distribution was normalized by the equivalent (log$_2$-transformed) distribution of non-modified ions using the R package preprocessCore[70]. To evaluate the statistical significance of the change in protein abundance, a linear model (adjusted on peptides and biological replicates) was performed, and a *t*-test was applied on the fold-change estimate. The *p*-values were then corrected for multiple testing using the Benjamini-Hochberg procedure. To represent the data with a heatmap, the intensities of the peptides were first transformed using the Z-score. Indeed, since for a given modification, the intensities are not comparable according to the peptide or the charge state, we choose to transform them. Finally, we aggregated the different values (using the mean) by replicates for each modification. This permits to obtain only one value by replicate, for each modification.

## RNA sequencing and analysis

**Total RNA sample preparation and sequencing.** For the comparison of 37 °C and 32 °C (Fig. 2a), P. *falciparum* parasites (strain NF54) were tightly synchronized by plasmion-sorbitol in a 6 h window, diluted to 2% parasitemia and half of the culture was put at 32 °C. Parasites grown at 37 °C were collected at 36 h.p.i., while parasites grown at 32 °C were collected at 56 h.p.i., which corresponded to the equivalent developmental stage (Supplementary Fig. 2h,i). For the LDH-glmS cell line, parasites were synchronized by plasmion/sorbitol to a 6 h window and diluted to 2% parasitemia. At 12 h.p.i., the culture was split and glucosamine (2.5 mM final concentration) was added to half of the culture. Parasites were collected at 36 h.p.i. (Supplementary Fig. 4b). Late stage *P. falciparum* Nico-glmS parasites were split and glucosamine (2.5 mM final concentration) was added to half of the culture. After 6 h, the parasites were synchronized by a plasmion/sorbitol in a 6 h window and diluted to 2% parasitemia. Parasites were collected at 36 h.p.i. (Supplementary Fig. 4b). For the comparison of *P. falciparum* wild-type (strain 3D7) and Sir2a-KO parasites, cells were synchronized by plasmion/sorbitol to a 6 h window and diluted to 2% parasitemia. Parasites were collected at 36 h.p.i. (Supplementary Fig. 4b).

For all asexual stages and the stage V gametocytes (*P. falciparum* strain NF54, Fig. 1a), parasites were lysed with 0.075% saponin in DPBS at 37 °C (or 32 °C for the cells cultivated at this temperature). The parasite cell pellet was washed once with DPBS and then resuspended in 700 μl QIAzol reagent (Qiagen # 79306). Total RNA was extracted using the Direct-zol RNA Microprep (Zymo # R2060), including an on-column DNase I digestion according to the manufacturer's protocol.

Midgut oocysts (8 days post bloodmeal) and salivary gland (21 days post bloodmeal) sporozoites were obtained by dissecting *Anopheles stephensi* mosquitoes infected with *P. falciparum* (strain NF54, Fig. 1a). Dissected midguts and salivary glands were placed in 250 mL of PBS (between 50 and 300 mosquitoes per replicate) and lysed by adding 750 mL of TRIzol-LS (Thermo Fisher # 10296010) followed by vigorous pipetting. Total RNA was obtained by chloroform extraction followed by isopropanol precipitation. The precipitated RNA was washed twice with 75 % ethanol and resuspended in RNase-free water.

All sequencing Libraries were prepared with the NEBNext Ultra II Directional RNA Library Prep Kit (NEB # E7760S) and sequenced (150 bp paired-end) on the Illumina NextSeq 500 platform.

**mRNA sample preparation and sequencing.** *P. falciparum* (strain NF54) parasites were synchronized by plasmion/sorbitol to a 6 h window. The culture was split after synchronization and one half incubated at 32 °C. The cells were collected at 15 h.p.i. (Supplementary Fig. 3e, f). Cells were lysed and RNA was extracted as described above. mRNA was enriched using the Dynabeads mRNA Purification Kit (Thermo Fisher # 61006). Library preparation and sequencing was performed as for the total RNA sequencing samples (see above).

**RNA-seq data processing and analysis.** Stringent read alignment filtering requires the removal of reads that align to multiple regions in the reference genome. As such, these approaches generally discard reads that map equally well to the near identical A1/2 and S2/3 loci (Supplementary Data 1). In contrast, also reporting non-unique (i.e. multi-mapping reads/secondary alignments) leads to an overestimation of the true abundance of transcripts that can originate from >1 near-identical loci[19,71]. To be able to maintain stringent alignment filtering (i.e. only reporting unique read alignments) and at the same time accurately measure rRNA abundance of A- and S-type rRNA, the A1 and S3 rDNA loci of the *P. falciparum* reference genome (PlasmoDB, version 64[10,72]) were masked from the 3′ end of the upstream gene to the 5′ end of the downstream gene using bedtools 'maskfasta'[73], leaving only one mappable copy of each rDNA type (i.e. A2 and S2) that are referred to as A1/2 and S2/3 throughout the study. A rDNA locus of *An. stephensi* was added as separate chromosome to the masked genome of *P. falciparum* and used as mapping reference for all samples originating from infected mosquitoes (i.e. oocysts and sporozoites).

For all RNA-seq experiments, raw sequencing data were base-called and demultiplexed with bcl2fastq and sequencing adapters were trimmed using trimmomatic[74]. Trimmed reads were aligned to the masked *P. falciparum* genome (see above) using STAR[75] with default options and option '--outFilterMultimapNmax 1'. Optical duplicates were removed using samtools 'fixmate' and 'markdup' and only alignments with both mates were retained (samtools view -f 0 × 2).

For the mRNA-seq experiment (Fig. 2b, Supplementary Data 6), gene counts were calculated from the filtered alignments using htseq-count[76] with options '-t exon -s reverse -r pos'. Fragments per kilobase of exon per one million mapped reads (FPKM) values were calculated and differential gene expression analysis was performed in R using DESeq2[77].

The mRNA transcriptome-based calculation of developmental age for parasites grown 37 °C and 32 °C (Fig. 2b, Supplementary Fig. 3e) was calculated in R using the method developed in Lemieux et al.[78] and using the transcriptome of Bozdech et al.[4] as baseline reference.

Approximation of the developmental age by comparison to scRNA[6] (Supplementary Fig. 3f) was calculated as follows: For each gene, the average FPKM across all three replicates was calculated and the expression of all genes was correlated to each individual transcriptome of Dogga et al.[6] in R using cor(method = "pearson"). Resulting Pearson $R^2$ values were plotted over the UMAP1 and UMAP3 coordinates of each individual cell in R using ggplot2[79].

To stringently align reads to the different rDNA loci, reads were aligned using STAR with default options except for '--alignEndsType EndToEnd', ' --scoreDelOpen -100' and '--scoreInsOpen -100' and subsequently filtered as above. To further remove misalignments among non-masked rDNA loci, only alignments with one or less mismatches per read pair (samtools view -e '[nM] <=1') and an alignment length ≥ 120 bp were retained. FPKM values and fold-changes of rRNA transcripts were calculated in R[80] (Supplementary Data 1). To quantify the change of contribution of transcription from A1 and A2 loci to the total pool of A-type 28S rRNA transcripts, filtered alignments at the A2 rDNA locus were visualized using the Integrative Genomics Viewer[79]. The frequency of alignments containing single nucleotide variants (SNVs) at the specific positions that differ between the A1 and A2 loci (i.e. reads originating from the A1 locus mapping to the A2 locus was calculated) was calculated. Fold-changes of SNV abundance were calculated relative to the asexual stage sample (Supplementary Fig. 1b).

For rRNA expression during midgut development (Supplementary Fig. 7h), the raw fastq files from Mohammed et al.[58] were mapped to the masked reference genome that includes a rDNA locus of *An. stephensi*. Alignments were filtered as described above for the total RNA sequencing samples. FKPM values for the calculation of relative contributions of each rDNA type were calculated in R using DESeq2[77].

**Chromatin Immunoprecipitation.** In general, we performed two independent biological replicates for all ChIP-seq experiments presented in this study.

**_P. falciparum_ chromatin preparation.** HMGB1-HA and HMGB2-HA cell lines were synchronized by plasmion/sorbitol to a 6 h window. After the synchronization, half of the culture was incubated at 32 °C. Parasites cultivated at 37 °C were collected at 36 h.p.i., while parasites incubated at 32 °C were collected at 56 h.p.i., which corresponded to an equivalent developmental stage (Supplementary Fig. 2h,i, 3a). For histone PTM ChIP-seq, wild-type *P. falciparum* (strain 3D7) and Sir2a-KO cell lines[41] were synchronized by plasmion/sorbitol to a 6 h window and collected at 16 h.p.i.

Chromatin Immunoprecipitation was performed as described previously[81]. Briefly, synchronized parasite cultures were lysed with saponin (0.075% in DPBS), washed with DPBS at 37 °C (or 32 °C for parasites cultivated at this temperature) and resuspended in DPBS at 25 °C. For HMGB1-HA and HMGB2-HA samples, parasites were first cross-linked for 20 min by adding glycolbis(succinimidylsuccinate) (EGS) to a final concentration of 1.5 mM. All samples (pre-cross-linked or not) were cross-linked for 10 min by adding methanol-free formaldehyde (Thermo Fisher # 28908) to 1% final concentration with gentle agitation. The cross-linking reaction was quenched by adding 2.5 M glycine to a final concentration of 0.125 M and incubated at RT for another 5 min. Parasites were centrifuged (3250 g, 5 min, 4 °C), washed with DPBS, snap-frozen, and stored at −80 °C until further use.

**Yeast chromatin preparation.** A culture of *Saccharomyces cerevisiae* was grown overnight in yeast peptide dextrose media (Bacteriological peptone 20 g/L, Glucose 20 g/L, Yeast extract 10 g/L) at 30 °C with agitation (300 rpm) to reach the stationary phase. The optical density (OD) was measured and the culture was diluted to an OD of 0.4. Cells were collected during the exponential phase at OD = 0.8 by centrifugation (5 min, 30 °C, 2,500 g) and crosslinking was performed by resuspending the cells in 2% formaldehyde and incubating at 25 °C for 5 min with rotation. 3 M Tris was added to a final concentration of 1.5 M to quench the crosslinking reaction, and the mixture was rotated at 25 °C for 1 min. After centrifugation (3 min, 4 °C, 2,500 g), the pellets were washed once with ice-cold PBS, and once with ddH₂O. After centrifugation (20 s, 4 °C, 3,300 g), the supernatant was removed and the pellets were snap-frozen and stored at -80 °C.

Cell pellets were thawed on ice and resuspended in 1X yeast ChIP lysis buffer (50 mM Hepes-KOH pH 8, 140 mM NaCl, 1 mM EDTA, 1% Triton X100, 0.1% sodium deoxycholate, 1 mM PMSF and 1X protease inhibitors). The cell walls were disrupted using a Beads Beater with zirconium beads and samples were recovered by centrifugation. The chromatin was sheared by sonication (8 cycles '30 s ON / 30 s OFF' in a Diagenode Pico BioRuptor) and centrifuged (20 min, 21,130 g, 4 °C). The supernatant was transferred to a new tube, aliquoted and stored at -80 °C. Yeast chromatin was mixed with *P. falciparum* chromatin as described below.

**Immunoprecipitation.** Each sample was resuspended in 2 ml nuclear extraction buffer (10 mM HEPES pH 8, 10 mM KCl, 0.1 mM EDTA pH 8, and complete protease inhibitor (PI, Roche no. 11836170001)). IGEPAL CA-630 was added to a final concentration of 0.25% and the cells were lysed with a prechilled douncer homogenizer (200 strokes). The nuclei were pelleted by centrifugation (13,500 g, 10 min, 4 °C), and the supernatant (cytoplasmic fraction) was removed while the pellet was resuspended in 1.8 mL of cold SDS Lysis buffer (50 mM Tris–HCl pH 8, 10 mM EDTA pH 8, 1% SDS, PI). Chromatin was sheared to ~250 bp fragments by 10 sonication cycles (once cycle: 30 s ON / 30 s OFF) using a Diagenode Pico Bioruptor. Insoluble materials were removed by centrifugation (10 min at 13,500 g at 4 °C) and DNA concentration was measured by Qubit.

For the wild-type and Sir2a-KO histone-PTM ChIP-seq experiments, *P. falciparum* and yeast chromatin was mixed at the following ratios (*P. falciparum:yeast*): H4K16ac: 2.5:1; H3K9ac, H3K14ac, H4K12ac: 1:1.

100 ng of DNA was stored as input sample at -20 °C for later processing (see below). For each immunoprecipitation, 25 µl of Protein G Dynabeads (Invitrogen 10004D) were pre-washed twice with 1 ml ChIP dilution buffer (16.7 mM Tris–HCl pH 8, 150 mM NaCl, 1.2 mM EDTA pH 8, 1% Triton X-100, 0.01% SDS) and resuspended in 1 ml ChIP dilution buffer. For HMGB1-HA and HMGB2-HA samples, 1 µg of anti-HA (Abcam # ab9110) was added to the beads and incubated at 4 °C for 2 h with rotation. For the wild-type and Sir2a-KO samples, 1 µg of either anti-H3K9ac (Merck # 07-532), anti-H3K14ac (Abcam # ab52946), anti-H4K12ac (Active Motif # 39066) or anti-H4K16ac (Merck # 07-329) was added and incubated at 4 °C for 2 h with rotation. The beads were washed twice with 1 ml ChIP dilution buffer and 1 µg of chromatin lysate was diluted 10-fold in ChIP dilution buffer, added to the bead-antibody complexes and incubated at 4 °C overnight with constant rotation.

Following immunoprecipitation, the beads were washed sequentially in 1 ml of low salt wash buffer (20 mM Tris–HCl pH 8, 150 mM NaCl, 2 mM EDTA pH 8, 1% Triton X-100, 0.1% SDS), High salt wash buffer (20 mM Tris–HCl pH 8, 500 mM NaCl, 2 mM EDTA pH 8, 1% Triton X-100, 0.1% SDS), LiCl wash buffer (10 mM Tris–HCl pH 8, 250 mM LiCl, 1 mM EDTA pH 8, 0.5% IGEPAL CA-630, 0.5% sodium deoxycholate) and TE wash buffer (10 mM Tris–HCl pH 8, 1 mM EDTA pH 8). For each wash, the beads were rotated for 5 min at 4 °C except for the last TE wash, which was performed at room temperature (RT). The beads were then resuspended in 200 µl of elution buffer (50 mM Tris–HCl pH 8 10 mM EDTA pH 8.0, 1% SDS) and protein-DNA complexes were eluted by 30 min incubation at 65 °C. The beads were collected on a magnetic rack and the supernatant transferred to a new tube.

**DNA purification and library preparation.** The immunoprecipitated DNA and the input DNA were reverse-crosslinked by incubating the samples at 65 °C for 10 h. 200 µl of TE buffer were added and the samples were treated with RNase A (0.2 mg/mL final concentration) for 2 h at 37 °C, followed by a 2 h incubation at 55 °C with proteinase K (0.2 mg/mL final concentration). DNA was purified by adding 400 µl phenol:chloroform:isoamyl alcohol, and phases were separated by centrifugation (13,500 g, 10 min, 4 °C) after vigorous vortexing. 16 µl of 5 M NaCl (200 mM final concentration) and 30 µg glycogen were added to the aqueous phase, and DNA was precipitated by adding 800 µl 100% EtOH 4 °C and incubating for 30 min at −20 °C. DNA was pelleted by centrifugation (20,000 g, 10 min, 4 °C) and washed with 500 µl 80% EtOH at 4 °C. After centrifugation, the DNA pellet was air-dried and resuspended in 30 µl of 10 mM Tris–HCl pH 8.0. Libraries were prepared with the NEBNext Ultra II DNA Library Prep Kit (NEB E7645S) and sequenced (150 bp paired end) on the NextSeq 500 platform (Illumina).

**ChIP-seq data processing and analysis.** Raw read pre-processing and adapter trimming were performed using bcl2fastq and trimmomatic[74] as described above for RNA-seq samples. Trimmed reads were aligned to the masked genome (see above) using bowtie2[82] with settings '--no-mixed --no-discordant --end-to-end --sensitive'. Alignments were subsequently filtered for PCR duplicates using samtools[83] 'fixmate' and 'markdup' and only alignments with a mapping quality ≥ 30 were retained (samtools view -q 30).

To calculate differences in HMGB1 occupancy at rDNA loci at 32 °C and 37 °C, IP/input enrichments were first calculated for both conditions and replicates independently using macs2[84] 'callpeak' with options '--nomodel --extsize 120'. The resulting tag numbers, pileup and lambda files for the IP and input samples of each replicate and condition were then used to calculate significant enrichment differences using macs2 'bdgdiff '. For all ChIP-seq experiments, coverage tracks representing the ratio of IP over Input were generated using deeptool's[85] 'bamCompare' with options '-bs 10 --scaleFactorsMethod None --operation ratio --normalizeUsing CPM'. IP/input coverage tracks were visualized using the Integrative Genomics Viewer[86].

For the analysis of histone PTMs in the wild-type[3D7] and Sir2a-KO parasites, raw sequencing reads were pre-processed as described above. The reads were then mapped to the masked *P. falciparum* genome (PlasmoDB v64[10,72]) using tinymapper (https://github.com/js2264/tinyMapper), with option '--mode ChIP' and using the *S. cerevisiae* S288C genome (version R64) as calibration reference. Only correctly paired reads with a mapping quality ≥ 30 were retained (bowtie2 option '-f 0 × 001 -f 0 × 002' -q 30) for downstream analysis within tinymapper. Significant peaks for each replicate and histone PTM were identified within tinymapper using macs2 and by using the input sample as background. To calculate quantitative enrichment values for each PTM, for each sample and replicate, the number of ChIP reads mapping to the *P. falciparum* genome (in counts per million [CPM]) was normalized to the total number of ChIP reads mapping to the *S. cerevisiae* calibration genome (Supplementary Data 10), allowing for the normalization of absolute peak enrichment (i.e. peak height) between samples (Supplementary Fig. 7e). Overlapping consensus peaks between replicates were identified using bedtool's 'intersect'[73] (Supplementary Data 10). For the comparison of 6 mA densities (Fig. 3f), first the absolute fold-change at the peak summit was calculated using the calibrated enrichment values for all significant peaks identified in the Sir2a-KO samples (i.e. calibrated enrichment Sir2a-KO/calibrated enrichment wild-type[3D7]). For the 100 peaks showing the highest and lowest change in peak enrichment, 6 mA densities identified by Sir2a-madID (see below) were calculated within the peak region. Of note, even peaks with the lowest change of absolute peak enrichment feature a fold-change of Sir2a-KO/wild-type[3D7] > 1 (Supplementary Data 10).

The raw ChIP and input fastq files for HP1[23] were retrieved from NCBI SRA (ChIP: SRR12281320; Input: SRR12281322) and processed as described above. The raw fastq and gDNA files for ATAC (assay for transposable accessible chromatin) sequencing data corresponding to 35 h.p.i. were retrieved from NCBI SRA (ATAC: SRR6055333; gDNA control: SRR6055335) and mapped to the masked genome as decribed above for ChIP-seq experiments. Read pairs with an insert size between 50−150 bp[21] were extracted using samtools 'view'[83] and the ration between the ATAC sample and gDNA control was calculated using deeptool's bamCompare[85]. All data were visualized using the Integrative Genomics Viewer[86].

**Micro-C analysis.** Post-processed, normalized matrices for the 36 h.p.i. timepoint were retrieved from Singh et al.[25]. Importantly, only alignments with a mapping quality (MAPQ) ≥ 30 were retained during the generation of interaction matrices. This ensures that reads that map equally well to either A-type or S2/3-type locus are removed and

otherwise would inflate the interaction frequencies between these loci. Intra- and interchromosomal interactions (Fig. 1g, Supplementary Fig. 1e) were plotted using hicExplorer's 'hicPlotmatrix'[87] at 5 kb resolution, except intrachromosomal interactions of Pf3D7_01_v3 which were plotted at 2 kb resolution. Intrachromosomal, $\log_2$-scaled observed/expected interaction frequency ratios (Supplementary Fig. 1e) were visualized using juicebox[88].

**Annotation of PolI factors.** For the annotation of Polymerase I-related factors in *P. falciparum*, proteins in human, Arabidopsis and yeast with GO term annotations related to PolI activity were retrieved from QuickGO[89] (Supplementary Data 2). Corresponding protein sequences were searched using BLASTp[90] against the *P. falciparum* proteome (PlasmoDB, v64[10,72]) with options '-evalue 1e-3 -max_target_seqs 2' and only hits with a SwissProt annotation was retained. The identified *P. falciparum* proteins were then used in a reciprocal BLASTp search against the model organism protein sequences and only the best hit was kept (option '-max_target_seqs 1') (Supplementary Data 2).

## Metabolomics

**Sample collection.** Asexual blood-stage *P. falciparum* parasites (wild-type[NF54], LDH-glmS and Nico-glmS cell lines) were tightly synchronized by plasmion-sorbitol in a 6 h window, iRBCs were concentrated by plasmagel flotation at 33 h.p.i., resuspended at $2.5 \times 10^7$ cells/mL in cell culture media and incubated in 12-well plates before collection at 40hpi. For LDH_GlmS, glucosamine (2.5 mM final concentration) was added at 16 h.p.i. to half of the culture, to obtain a total of 24 h of glucosamine treatment. For Nico-glmS, glucosamine (2.5 mM final concentration) was added at 6 h before synchronization to half of the culture, to obtain a total of 48 h of glucosamine treatment. For the 32 °C vs 37 °C temperature experiment, half of the culture was put at 32 °C after the plasmion at 33 h.p.i. Due to slower growth, the cells incubated at 32 °C were collected 2 h later, at 42 h.p.i. (Supplementary Fig. 2h).

For each condition, 6–10 replicates of $5 \times 10^{-7}$ cells were collected: the cells were transferred to a 2 ml tube, centrifuged at 14,000 g for 30 s at 37 °C, washed with 1 ml of ice-cold DPBS, centrifuged at 14,000 g for 30 s at 4 °C and the pellet was snap-frozen in liquid nitrogen and stored at -80 °C.

**LC-MS determination of energy carrier metabolites.** Frozen cell pellets were processed following an adjusted protocol targeting energy carriers such as NAD/NADH and NADP/NADPH. This method was extended by including polarity switching and additional metabolites of interest. Briefly, cell pellets were extracted on ice using 250 µl cooled extraction buffer (Acetonitrile: MeOH: 15 mM ammonium acetate in $H_2O$ (3:1:1), pH 10). Subsequently, samples were sonicated to ensure complete disruption of all cells using a sonication bath (Transsonic 460, Elma) for 5 min at the highest frequency on ice. Afterwards, samples were centrifuged for 15 min at 4 °C and 13,000 g, and the resulting supernatant was transferred to a new LC-MS grade autosampler vial and immediately frozen at -80 °C if instrument was not directly available for measuring.

For metabolite separation and detection, an ACQUITY I-class PLUS UPLC system (Waters) coupled to a QTRAP 6500+ (AB SCIEX) mass spectrometer with electrospray ionization (ESI) source was used. In detail, metabolites were separated on an ACQUITY Premier BEH Amide Vanguard Fit column (100 mm × 2.1 mm, 1.7 µm, Waters) with constant column temperature of 35 °C. Separation of NAD/NADH, NADP/NADPH and additional energy carriers was achieved using mobile phase A (50/50 ACN/water with 5 mM ammonium acetate and 0.04% ammonium hydroxide; pH 10) and mobile phase B (90/10 ACN/water with 5 mM ammonium acetate and 0.04% ammonium hydroxide), following a gradient of the A/B phase ratio (0.5 min 5%/95% at 0.4 ml/min, 4.5 min 90%/10% at 0.35 ml/min, 5 min 100%/0% at 0.3 ml/

min and 5 min 5%/95% at 0.4 ml/min). Data acquisition was performed using Analyst 1.7.2 (AB SCIEX) and processed using the OS software suite 2.0.0 (AB SCIEX).

Areas under the curve (auc) for each metabolite were normalized to $1 \times 10^6$ cells and NAD/NAM ratios were calculated for each replicate. To estimate relative changes of NAD/NAM, all replicates were normalized to the average of the wild-type/untreated sample. Normal distribution of the data were estimated in R using functions shapiro.test() and *p*-values were computed using an unpaired Welch Two Sample *t*-test using function *t*.test(paired = F).

**ONT sampling and library preparation for HMGB1 integration check.** Late stage HMGB1-HA parasites were collected by saponin lysis (0.075% saponin in DPBS) at 37 °C. The parasite cell pellet was washed twice with ice-cold DPBS, snap-frozen and stored at -80 °C. gDNA was extracted using the Qiagen DNeasy Blood & Tissue Kit (Qiagen # 69504). The gDNA was further purified by phenol/chloroform extraction and ethanol precipitation. Briefly, 200 µl of Phenol/Chloroform/Isoamyl Alcohol (25:24:1) was added to 200 µl of gDNA in elution buffer. The samples were vortexed and centrifuged (4 °C, 21,000 g, 5 min). 180 µl of the upper phase was transferred to a new tube and an equal volume of Phenol/Chloroform/Isoamyl Alcohol was added to repeat the extraction. After agitation and centrifugation, the upper phase was transferred to a new tube and 1/10 volume of 3 M sodium acetate was added, together with 1 µl of 10 mg/ml glycogen and two volumes of 100% ethanol. The mixture was centrifuged (4 °C, 21,000 g, 20 min) and the resulting DNA pellet was washed with 80% ethanol, air-dried and resuspended in ddH2O. Libraries were prepared using the SQK-RAD004 kit and sequencing was performed using a FLO-MIN106 flow cell on a MinION sequencer. Real-time basecalling was performed using the fast model in Guppy (v 6.5.7) implemented in MinKNOW (v23.04.5). The expected pSLI plasmid integration was manually integrated at the HMGB1 locus of the reference *P. falciparum* (PlasmoDB v64[10,72]) genome and annotation. Raw fastq reads were mapped to the modified genome sequence using minimap2[91] with option '-x map-ont' and the resulting alignments visualized using the Integrative Genomics Viewer[86].

**Native cDNA sequencing.** To annotate the upstream non-coding RNAs co-expressed with S2/3-type rRNA, wild-type[NF54] parasites were grown at 32 °C and tightly synchronized using plasmion/sorbitol to a 6 h window. Parasites were collected at 40 h.p.i. and total RNA was extracted using phenol-chloroform followed by ethanol precipitation. Native cDNA sequencing libraries were prepared using the SQK-DCS109 native cDNA kit and sequencing was performed with a FLO-MIN106D (R9.4.1) flow cell on a MinION sequencer. Real-time base-calling was performed using the fast model in Guppy (v 6.5.7) implemented in MinKNOW (v23.04.5). Raw fastq reads were mapped to the unmasked *P. falciparum* reference genome (PlasmoDB v64[10,72]) using minimap2[91]. Alignments with a MAPQ score ≥ 60 were filtered using samtools[83] 'view' (option -q 60) and visualized using the Integrative Genomics Viewer[86].

**ONT sampling and library preparation for madID.** For NLS-madID, cells were synchronized by plasmion/sorbitol to a 6 h window. To dimerize and activate the Cre recombinase, rapamacin was added (200 nM final concentration) after the sorbitol, and cells were collected at 40 h.p.i. Successful excision of the yDHODH locus was confirmed by PCR (Supplementary Fig. 6h). For Sir2a-madID, the cell line was synchronized by plasmion/sorbitol to a 6 h window and collected at 40 h.p.i. Red blood cells were lysed with 0.075% saponin in DPBS at 37 °C. The parasite cell pellet was washed twice with ice-cold DPBS, snap-frozen and stored at -80 °C. gDNA was extracted using the Qiagen DNeasy Blood & Tissue Kit (Qiagen # 69504). gDNA was purified by treatment with RNase A (0.2 mg/mL final concentration) for 2 h at

37 °C, followed by a 2 h incubation at 55 °C with proteinase K (0.2 mg/mL final concentration). The obtained gDNA was further purified by phenol/chloroform extraction and ethanol precipitation. Libraries were prepared using the native barcoding SQK-NBD114.96 kit using the recommendations for long gDNA fragments. Sequencing was performed using a MinION flow cell (R10.4.1) on a MinION sequencer.

**Identification of madID-modified DNA sites.** Raw pod5 files were merged using pod5 'merge' (https://github.com/nanoporetech/pod5-file-format). Basecalling and mapping to the unmasked reference genome of *P. falciparum* was performed in a single step using dorado basecaller (v0.8.2, https://github.com/nanoporetech/dorado) and minimap2[91], respectively, using options '--emit-moves --emit-sam --kit-name SQK-NBD114-96 --reference PlasmoDB-64_Pfalciparum3D7_Genome.fasta sup@v5.0.0'. Model'dna_r10.4.1_e8.2_400bps_sup@v5.0.0' was used for base calling. Individual samples were demultiplexed using dorado 'demux' (v0.8.2) and alignments with a MAPQ score ≥ 60 were filtered using samtools 'view' (option -q 60). For each sample (i.e. gDNA, NLS-madID and Sir2a-madID), raw signal alignments were performed using uncalled4[92] 'align' with model 'dna_r10.4.1_400bps_9mer'. For each genomic position and sample, the normalized mean read signal current was calculated using uncalled4[92] 'refstats' with options '--stats mean --layers dtw.current'. Genomic positions with a normalized mean read signal current difference ≥2 between NLS-madID vs gDNA (i.e. 'madID', Fig. 3) and Sir2a-madID vs gDNA (i.e. 'Sir2a-madID', Fig. 3) were retained as putatively modified sites. For genome wide visualizations, the number of modified sites was normalized to the AT content of each bin (Fig. 3c, g, h, Supplementary Fig. 7a, f, g). Regions with a ≥ 10-fold enrichment between the Sir2a-madID and madID samples (Supplementary Data 9) were identified by merging adjacent bins (100 bp) that feature a 6 mA occupancies ≥ 10 in the Sir2a-madID sample using bedops '--merge'[93]. Fold-change enrichments among rDNA loci in the Sir2a-madID sample were calculated using the maximum 6 mA enrichment values in the upstream regions of each locus (Supplementary Data 11). For the comparison of modified sites in heterochromatic and euchromatic regions (Fig. 3d), the total number of modified sites was normalized to the length of each region (i.e. number of modified sites per kilobase). Location of heterochromatic and euchromatic regions was retrieved from Singh et al.[25] *P*-values for the comparison of 6mA densities in euchromatic and heterochromatic regions (Fig. 3d) and between peak regions (Fig. 3f) were computed in R using a Wilcoxon Rank Sum Test with function wilcoxon.test(paired = F).

**Dot blot.** The gDNA of NLS-madID (with and without rapamycin treatment) and Sir2a-madID was extracted as described above and denatured for 5 min at 65 °C. Serial dilutions were performed to blot 2.5 µl of sample corresponding to 100 ng, 10 ng and 1 ng of gDNA on a nylon membrane (Biodyne # 77016), and the membrane was air-dried for 10 min. The membrane was UV-crosslinked (125 mJ/cm² at 254 nM for 1 min) and blocked for 1 h in 5% milk in TBST (50 mM Tris, 150 mM NaCl, 0.05% Tween20, pH 7.5). The membrane was then incubated with anti-6mA antibody (Abcam # ab151230) diluted at 1/1,000 in blocking solution for 2 h at RT, washed three times with TBST for 5 min and incubated with HRP anti-rabbit (Sigma # NA934) diluted at 1/2500 in blocking solution for 1 h at RT. The membrane was washed three times with TBST for 5 min, developed using the SuperSignal West Pico chemiluminescent substrate (Thermo Fisher # 34580) and imaged with a ChemiDoc XRS+ (Bio-Rad).

**Reporting summary**
Further information on research design is available in the Nature Portfolio Reporting Summary linked to this article.

## Data availability
All raw Illumina sequencing reads and aligned ONT samples generated in this study have been deposited in NCBI BioProject database under accession code PRJNA1224397. An overview table of all samples with the respective SRA number is shown in Supplementary Data 14. ChIP-seq and madID tracks have been deposited in the Gene Expression Omnibus (GEO) database under accession code GSE290936 (madID) and GSE290639 (ChIP-seq). The mass spectrometry proteomics data have been deposited in the ProteomeXchange Consortium (http://proteomecentral.proteomexchange.org) via the PRIDE partner repository[94] with the dataset identifier PXD060142. Previously published data used in this manuscript are available under the following NCBI SRA accession numbers: HP1 ChIP (SRR12281320) and HP1 input (SRR12281322); ATAC-seq (SRR6055333) and ATAC gDNA control (SRR6055335). Micro-C data are available at the GEO database with accession GSE278141.

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

## Acknowledgements

We thank the Metabolomics Core Technology Platform of the Heidelberg University supported by the CellNetworks Core Technology Platform CCTP for metabolite analyses. We thank Christophe Chapard for providing help with yeast culturing and collection, Julien Guizetti for sharing plasmids for madID cloning and Patrick Poullet for the continuous development of myProMS used for proteomics analyses. The Photonic BioImaging platform gratefully acknowledge the kind financial support of the Institut Pasteur (Paris), the France–BioImaging infrastructure network supported by the French National Research Agency (ANR-10–INBS–04), and the Région Ile-de-France (program DIM-Malinf). Work in the laboratory of S.B. is supported by an ERC Starting Grant (# 947819) and baseline funding of the Institut Pasteur. J.M.B. is supported by ANR ANR-21-CE15-0002-02 ApiMORCing and ANR-21-CE15-0010-01 PlasmoVarOrg.

## Author contributions

S.B., J.M.B and J.E.C conceptualized the study. J.E.C performed all ChIP-seq and RNA-seq experiments with help from T.V. and G.D. B. L. carried out the proteomics experimental work, M.R. analysed proteomics data with help from D. L. M.B., G.P. and R.D.-S. performed and analyzed metabolomics experiments. S.M. carried out immunofluorescence experiments. J.E.C., J.M.B. and S.B. wrote the manuscript with input from all authors.

## Competing interests

The authors declare no competing interests.
