## [Transparent Peer Review file · Nature Communications]

A metabolism-chromatin axis promotes differential ribosomal RNA transcription in the human malaria parasite

Corresponding Author: Dr Sebastian Baumgarten

Version 0:

Reviewer comments:

Reviewer #1

(Remarks to the Author)

Please see the attached Word file with a figure.

Reviewer #2

(Remarks to the Author)

The submission from Couble and colleagues describes the molecular basis underlying the transition between alternative ribosomal RNA expression during differentiation by the human malaria parasite *Plasmodium falciparum*. Malaria parasites that infect a range of hosts all display a unique property of having rRNAs encoded by a handful of genomic loci rather than the large copy number arrays that are typical of eukaryotic organisms. Even more unusual, these organisms activate different rRNA genes when transitioning between their mammalian and mosquito hosts. The biological significance underlying these rRNA transitions is not known, however changes in glucose availability and temperature were previously shown to be important cues leading to changes in rRNA expression. Here the authors identify shifts in metabolism resulting in changes in NAD⁺ availability as a key contributor to this change in rRNA expression, and further show the importance of altered activity of the histone deacetylase Sir2a.

The authors provide a substantial amount of data to support their model and overall, the study is an important contribution to our understanding of this fascinating phenomenon. However, the manuscript would benefit from some additional clarity in several parts to further reinforce the authors' conclusions and to not overstate the effect that they document. Specific comments are below:

1. Much of the manuscript is focused on understanding the activation of expression of the S2/3 rDNA genes through changes in temperature or metabolic manipulations. Throughout the manuscript, this is shown as Fold Change in RNA. Several previously published studies have documented this phenomenon and shown changes in S2/S3 expression up to 30-50-fold (for example Fang et al, JBC 2004 and Samuel et al, Pathogens, 2024). Here the authors shown a consistent, although smaller change in expression. For example, Figure 2A displays an increase in S2/S3 expression of around 16-fold. Do the authors think the lower change here is because they limited the temperature drop to 32 degrees rather than the 27 degrees used in previous studies? Similarly, the manipulations of LDH levels, Sir2a levels or Nicotinamidase levels resulted in changes in expression by only 3-4-fold, much less than that observed previously. The authors note in the Discussion that "S-type rRNA do not reach the same absolute expression levels as they normally do in mosquito stages" and note the possible contribution of additional activation of the TCA cycle in mosquito stages (e.g. through α -ketoglutarate). However, it is not clear why the manipulations of Sir2a/LDH/Nicotinamidase do not meet the level of change shown in Figure 2A. Is it possible that these conditions are important for keeping the loci silent but are not sufficient to induce a fully active state, suggesting that an additional, temperature responsive activator is missing in these experiments? Any thoughts or clarifications would be useful.

2. The authors described the chromatin landscape at active and silent rDNA loci. This is a primary aspect of the paper. However, the data presented are not complete. For example, H3K9ac is noted as an important mark for transcriptionally

active chromatin and a peak is shown upstream of the active A2 locus (Figure 1E), however no data are presented for the A1 locus. Similarly, the absence of this mark is shown in supplemental Figure S1C for the S1, S3 and O2 loci, but S2 and O1 are not shown. This is despite the text stating that a peak of H3K9ac is not found “at any of the silent rDNA loci in asexually replicating cells.” Similarly, the distribution of HP1 and ATAC-seq accessibility obtained from previously published datasets omits the A1 and S2 loci. Is there a reason these data are not available or not included in the analysis shown in the paper? As currently written, the analysis appears incomplete.

3. The authors note that in response to a shift in temperature to 32 degrees, HMGB1 is recruited to the S3 locus (Figure 2B). This figure does indeed show a somewhat subtle, but clear increase in HMGB1 at 32 degrees. They then show very little association of HMGB1/2 at the silent loci in Figure S2G. Presumably the data in Figure S2G are from parasites grown at 37 degrees, although the Figure legend does not make this clear. More importantly, does the presence of HMGB1 or 2 increase at these loci when the temperature is decreased to 32 degrees? The text says mentions recruitment of HMGB1 to the S2/S3 loci at 32 degrees and cites both Figures 2B (which shows the S3 locus) and Figure S2F (which I think should be S2G, and which only shows the absence of HMGB1 and 2 at four loci). Thus, the data are somewhat incomplete and do not directly coincide with what is stated in the text. Similarly, the noncoding “truRNAs” upstream of the S3 locus (Figure 2B) have also been reported to be upregulated in response to a temperature drop (Sharma et al, Genes, 2023 and Samuel, Pathogens, 2024), however there appears to be no change in HMGB1 occupancy. Thus, the conclusions are somewhat muddled by the lack of complete data or a bit inconsistent with the stated model. A more complete explanation accommodating all the available data would be helpful.

4. The authors knocked down Nicotinamidase expression and noted a de-repression of previously silent S-type rRNA, which on line 172 they note is “similar to that which takes place upon growth at 32 degrees.” However, as mentioned in point 1 above, it should be noted by the authors here that the change observed is a fraction of what was observed at 32 degrees, a little over 3-fold as compared to the ~16-fold observed by the temperature shift. A similar magnitude effect is observed in the LDH1 and Sir2a knockdowns.

5. The authors mapped the interactions of Sir2a throughout the genome and noted that it appears to be preferentially associated with euchromatic regions of the genome, suggesting a possible role in modulating histone acetylation in transcriptionally permissive chromatin regions (lines 251-158). They then note the “exception” of var introns and mention var intron-derived transcripts. The logic here is not clear. If transcripts come from var-introns (and thus they are transcriptionally permissive), then the association of Sir2a with var introns would seem to be consistent with its association with euchromatic (and transcriptionally permissive) regions elsewhere in the genome. It is not clear why the authors consider this an “exception”.

Reviewer #3

(Remarks to the Author)

With the presented article “A metabolism-chromatin axis promotes ribosome heterogeneity in the human malaria parasite” the authors provide an intriguing and timely manuscript. Using a combination of forward genetics, metabolomics and state-of-the-art chromatin/epigenetics tools and experiments, the authors provide compelling mechanistic insights how a link between the metabolic state of the parasite and the epigenetic machinery controlling gene silencing ensure the stage specific expression of divergent ribosomes in *Plasmodium falciparum*.

The authors achieve this by providing insights on how the epigenetic state/transcriptional repression of the different rRNA genes is influenced by culturing the parasites at lower temperature and then link these changes to the NAD⁺/NAM levels in the parasite using targeted metabolomics and conditional KO parasites for the nicotinamidase NICO, a key enzyme in the NAD⁺ metabolism. Using an additional conditional KO line for the LDH1 protein, the authors are further able to recapitulate the decreased temperature phenotype / NICO knockdown phenotype, indicating how the shift from aerobic glycolysis to more oxidative phosphorylation during the build-up to the human to mosquito transmission could be the driving force behind this metabolic control of rRNA expression. Finally, the authors can present the histone deacetylase Sir2a as likely direct target of the changing NAD⁺/NAM ratios and key player in “translating” these changes in NAD⁺ levels into transcriptional regulation of the rRNA genes by changes in chromatin modifications, namely histone acetylation. While the presented effect of changes in the glucose metabolism on rDNA transcription and the involvement of Sir2a as a key regulator of rDNA transcription have been investigated before (as cited by the authors), the authors do a nice job in connecting these observations and add significant new mechanistic insights using the NICO, LDH1 and Sir2a KO parasites lines. The methodology is sound; all experiments have been conducted with the appropriate number of biological replicates and controls. In addition, the used methods of Micro-C and Sir2a-madID sequencing go beyond the “state-of-the-art” in the field and nicely showcase the adaption of these methods for molecular research in *P. falciparum*.

In summary, the presented manuscript will be highly significant to the field and nicely complements recent insights in the regulatory link between metabolism and the epigenetic regulation of sexual development and immune evasion in *P. falciparum* (see e.g. Brancucci NMB et al., 2017, Schneider VM et al., 2023) At the same time, the link between gene regulatory/epigenetic processes and metabolism/metabolic shifts are current topics in model organisms as well as medically relevant fields such as cancer epigenetics. The presentation of possible common principles or markedly different adaptations governing these processes from model organisms to diseases phenotypes and pathogenic single celled organisms like *P. falciparum*, will thus also be of significant interest to a broader audience.

Minor comments:

In general, I would appreciate, if the chromosome numbers would be added in addition to the genomic coordinates in the ChIP / madIDseq graphs. This would make orientation in the dataset a bit easier.

Line 135 onwards:

“To identify proteins potentially involved in the structuring of the rDNA chromatin environment and/or recruitment of Poll to active rDNA loci, we annotated the Poll machinery in *P. falciparum*. This search identified high-mobility group proteins 1 (HMGB1 PF3D7_1202900) and 2 (HMGB2, PF3D7_0817900) as putative homologs of human nucleolar transcription factor 1 (Table S2). This protein is required for Poll recruitment and subsequent remodeling of active rDNA chromatin and rRNA transcription²³. Epitope-tagging of the two proteins with 3x hemagglutinin (3xHA) (Figure S2A-E) followed by ChIP-seq revealed that they specifically associate with the active, but not the silent rDNA loci (Figure 1H,I Figure S2F,G, Table S3), providing evidence for an orthogonal role of these proteins similar to other eukaryotic nucleolar transcription factors.”

I would not define HMGB1/2 as homologs to UBF1. The presented blast hit is due to the presence of the HMG-box motif in UBF1. However, UBF1 is a bigger multidomain protein and not comparable to HMGB1/2 in *Pf*. A better comparison here might be the described role of yeast HMO1 in rDNA regulation (Huffines AK, 2024). I would thus suggest changing the wording in this paragraph.

A recent study characterized the HMGB1 protein in asexual stage of *P.f.* (Lu B, et al., Parasitology, 2021). In the study the authors generated a HMGB1 KO parasite line which showed no apparent growth defect or any remarkable changes in rDNA expression regulation (as far as this reviewer is aware). Do the authors have an idea how this fits with the proposed role of HMGB1 as nucleolar transcription factor?

Fig S7A:

Was the var intron peak the only discernible enrichment of Sir2A in the genome? I couldn't find a table or similar with possible Sir2A peaks. In case there were additional peaks detectable such a table overview would be interesting and could be added.

In summary I enjoyed reviewing the presented manuscript, congratulate the authors to a great study and would recommend publication with these minor revisions.

Best regards.

Version 1:

Reviewer comments:

Reviewer #1

(Remarks to the Author)
See the attached Word file

Reviewer #2

(Remarks to the Author)

The revised submission from Couble and colleagues provides an important contribution to our understanding of the transition between alternative ribosomal RNA expression during differentiation of malaria parasites. This is a fascinating phenomenon that has puzzled researchers for decades. By linking this process to specific metabolic changes, the manuscript provides key insights into how the parasites transition to the environment of the mosquito vector.

The initial version of the paper primarily suffered from a lack of clarity regarding how sequencing data for the different ribosomal RNA types were analyzed and presented (specifically the “masking” of specific loci) and the need for greater explanation regarding the level of de-repression observed in this experimental model as compared to actual rRNA expression levels in the mosquito. The authors have suitably addressed these concerns by expanding and clarifying the text and by modifying the figures appropriately. Overall, the data are now appropriately presented and interpreted. I have only a couple of very minor suggestions for the authors.

1. On line 53-54, the authors state “...distinct rRNA genes are assembled into possibly divergent ribosomes.....” This should probably be written “wherein the products of distinct rRNA genes are assembled into possibly divergent ribosomes...” Genes are not assembled into ribosomes, rRNAs are.

2. On line 83-84, the authors note “Thus, A2 and S2 will be representative of the A1/2 and S2/3 rDNA loci, respectively.” As mentioned above, this is an important clarification. However, some readers might be a bit confused since the first Figure shows all loci separately. Perhaps this sentence could be modified to say “Thus, when mapping sequencing reads, A2 and S2 will be representative of the A1/2 and S2/3 rDNA loci, respectively.”

Thank you again for submitting your manuscript "A metabolism-chromatin axis promotes ribosome heterogeneity in the human malaria parasite" to Nature Communications. We have now received reports from 3 reviewers and, after careful consideration, we have decided to invite a major revision of the manuscript.

As you will see from the reports copied below, the reviewers raise important concerns. We find that these concerns limit the strength of the study, and therefore we ask you to address them with additional work. Without substantial revisions, we will be unlikely to send the paper back to review. In particular, given concerns with the physiological relevance of your findings (reviewer #1) it won't be sufficient to clarify concerns related to marginal changes in transcript levels, chromatin markers, HMGB1 and 2 occupancy collectively raised by all reviewers. Instead, we expect additional experimental data reporting rRNA gene level data instead of grouping as currently done in your work, clarifying how manipulation of LDH, Sir2a, Nicotinamidase levels and temperature change transcript levels, the chromatin landscape, and HMGB1 and 2 occupancy. Please also clarify the rationale for testing 32°C and be more specific with your title in reflecting the major findings of your work.

If you feel that you are able to comprehensively address the reviewers' concerns, please provide a point-by-point response to these comments along with your revision. Please show all changes in the manuscript text file with track changes or colour highlighting. If you are unable to address specific reviewer requests or find any points invalid, please explain why in the point-by-point response.

Reviewer #1 (Remarks to the Author):

The Plasmodium parasite evolves several divergent copies of rRNA genes, which are localized in different chromosomes. Interestingly, the A-type (A1 and A2) rRNA shows ubiquitous expression in all the development stages, including human and mosquito host stages; however, the S-type (S1, S2, and S3) rRNA is specifically expressed in oocyst and sporozoite at the mosquito host, but not in the human host. The regulation and mechanism of mosquito stage-specific expression of S-type (S1, S2, and S3) rRNA remains elusive.

The study by Couble et al. revealed that the epigenetic signatures of rRNA transcription and found that rDNA silencing relies on aerobic glycolysis in the human host.

Disruption of NAD⁺ regeneration during lactate fermentation promotes rDNA de-repression and identify the sirtuin histone deacetylase Sir2a as the mediator between fluctuating NAD⁺ levels and a functional transcriptional outcome.

Major concerns:

> For the five copies of rRNA genes, the A type (A1 and A2) rRNA shows ubiquitous expression in all the developmental stages, including human and mosquito host. However, the S type (S1, S2, and S3) rRNA is specifically expressed in oocyst and sporozoite at the mosquito host, but not in the human host. The author called the no expression of S-type rRNA in human host as gene silencing or repression, and the oocyst- and sporozoite-stage specific expression of S-type rRNA as gene de-repression. During the parasite life cycle, lots of genes show stage-specific expression. It complicates the stage-specific gene expression as “gene silencing/repression and gene de-repression”.

The reviewer is correct that many genes are differentially expressed or completely silenced during the parasite lifecycle. We specifically chose to call the induction of silent rDNA transcription in the mosquito stages “de-repression” because we identified heterochromatin protein 1 (HP1) as a direct repressor located at these loci in their silent stage during asexual replication (Fig. 1f, Supplementary Fig. 1d). This contrasts with other silent genes during asexual replication that are located in transcriptionally permissive euchromatin across the central regions of the *P. falciparum* genome that are not actively repressed by HP1, but whose

activation relies on the stage-specific presence of an activating transcription factor (e.g. an AP2 protein). We clarified our language in this regard on lines **121ff** and **346ff** and **370ff**.

> The author investigated the effect of the metabolism and chromatin on the gene expression for both A-type (A1 and A2) and S-type (S1, S2, and S3) rRNA in the asexual blood stage in the human host. In the manuscript, no direct evidence showed the ribosome heterogeneity. It is not appropriate for the title “A metabolism-chromatin axis promotes ribosome heterogeneity in the human malaria”.

We agree with the reviewer’s critique and have changed the title to ‘A metabolism-chromatin axis promotes differential ribosomal RNA expression in the human malaria parasite’.

> In Fig 2A, the author investigated the transcript expression of A-type (A1 and A2) and S-type (S1, S2, and S3) rRNA in the asexual blood stage at 37°C and 32°C, respectively, and found that the S-type rRNA showed pronounced upregulation at 32°C. I checked the data of the normalized expression value in FPKM in Table S4 (see below). Compared to 37°C, the transcript amount of S1 rRNA increases from 65 to 254, and S2/S3 rRNA increases from 15 to 233. However, the upregulated level of S-type (S1, S2, and S3) rRNA is quite much lower than the level (30000-50000 in FPKM) of S-type (S1, S2, and S3) rRNA in oocyst and sporozoite. Even after a 10-fold upregulation at 32°C, it is still less than 1% of the physiological expression level (30000-50000 in FPKM) in the oocyst and sporozoite stages! I agree that the environmental temperature switch from 37 to 32°C has an effect on the expression of S-type rRNA, but it just plays a minor role. This change in transcript level is far from the real physiological change in the transcript level between the asexual blood stage and oocyst/sporozoite stage.

The reviewer is correct that none of our perturbations result in an upregulation of silent rRNA in asexual parasites that compares to that in the mosquito stages. In fact, asexually replicating parasites never encounter temperatures that are significantly lower than 37°C in the human host. As such, our study actually shows that temperature is not a primary factor for silent rRNA upregulation. However, silent rRNA upregulation in asexual parasites in response to artificially decreased temperatures in culture has been reported multiple times over many years (¹⁻⁴). This led us to use this setup at the beginning of our investigation (line **162ff, 360ff**) as an experimentally tractable *model* aiming to understand the mechanism of how silent rRNA can be

de-repressed/activated on a chromatin level. Indeed, even though we do not reach physiological levels of S-type de-repression, we still observe significant changes in chromatin signature at these rDNA loci upon temperature decrease. While it is not possible to currently perform these extensive genome-wide studies (e.g. ChIP-seq) in oocysts due to difficulties obtaining enough material, we believe that we would see even more pronounced chromatin changes in the real physiological context as S-type rRNA are up-regulated.

This temperature *model* also just represents the backdrop of our study as we continued to stepwise dissect the metabolic dependency of rRNA expression. We strongly believe that this link between the primary metabolic pathway (and the switch between them) and rRNA expression we present here is very much of physiological relevance (new Fig. 3i). We have now substantially added and revised this section on the reasoning of our experimental approach and the differential degrees of upregulation depending on the experimental setup on line **161ff**, **360ff**, and **384ff** of our manuscript (please see also below).

We also want to highlight that even a 50-fold increase observed in previous studies that used lower temperatures (¹⁻⁴) still falls far short of the S-type rRNA levels measured in the mosquito. Nonetheless, these studies equally used such an approach to measure S-type transcription. Hence, we believe that the consistent pattern of upregulation that we and others have found is equally important to the absolute level of S-type upregulation itself and is sufficient to decipher the molecular underpinnings of this upregulation (line **169ff** and **399ff**).

> Similar concern as above In Fig 2F, only a 2-4 fold upregulation of S-type (S1, S2, and S3) rRNA after knockdown of Nico gene. In Fig 2H, only a 2-4 fold upregulation of S-type (S1, S2, and S3) rRNA after knockdown of LDH gene. In Fig 2I, only a 2-2.5 fold upregulation of S-type (S1, S2, and S3) rRNA after knockout of Sir2a gene.

The reviewer is correct that none of the conditions (or those tested in previous studies investigating rRNA expression ¹⁻⁴) are sufficient to increase the levels of S-type rRNA to those measured in the mosquito. Overall, the goal was not to upregulate S-type rRNA expression in asexually replicating parasites and resolve ribosome heterogeneity pretending that those are equivalent to mosquito stages. We very much agree with the reviewer that those are very different developmental states. Our goal was to 1) show that S-type rRNA repression is a feature of asexually replicating parasites and 2) elucidate how this is dynamically regulated,

which is physiologically relevant to the asexually replicating parasite (see above). As such, our experimental model did not allow us to resolve ribosome heterogeneity (which was also not the goal, and we have also adjusted our title accordingly), but to define the states of active and silent chromatin states and use different perturbations to measure the underlying dynamics that lead to the consistent pattern of rRNA expression we find.

We have substantially extended the discussion in this regard on lines **360-403**. We describe the differences of S-type transcription levels between 1) the experimental model in asexual parasites and mosquitoes (line **367ff**), 2) previous studies and ours (line **360ff**), and 3) the additive effect temperature can have on rRNA homeostasis (line **384ff**) and what our model allowed us to do and not do (line **392ff**).

> In addition, lots of other genes were differentially expressed in the asexual blood stage at 32°C (see Fig 2C). The gene expression regulation by lower temperature is not specific to the S-type rRNA genes.

The reviewer is correct that multiple genes are up-/downregulated, including LDH1 and Nicotinamidase, as described in our work. We now include a genome-wide comparison of gene expression from the total RNA sequencing between 37°C and 32°C (new Supplementary Figure 3c), showing that silent rRNA transcripts (18S and 28S and the ncRNA upstream of S2) are among the most highly upregulated transcripts. We describe how our working hypothesis explains this phenotype on lines **346ff**.

> A1, A2, S1, S2, and S3 rRNA are five individual genes. In many figures (Fig 2A, Fig 2F, Fig 2H, Fig 2I, and several supplementary figures), A1 and A2 were merged as one group A1-2, while S2 and S3 were merged as one group S2-3. What is the reason? All the individual rRNA genes should be analyzed and presented individually.

As this is a concern raised by the reviewer again below and also by reviewer 2, we will explain our reasoning of 'merging' in detail here and have also expanded and clarified our approach on line **1045ff** of the revised manuscript.

Every distinct molecule sequenced in an Illumina run can only originate from one specific position of the physical genome. Short read aligners such as those used here (STAR for RNA-seq, bowtie2 for ChIP-seq) attempt to find and report this specific position with the highest

confidence possible (i.e. the MAPQ score reported as $Q = -10 \cdot \log_{10}(p)$, where p equals the probability that the reported alignment is not the true location from where the sequenced molecule originated). Thus, the most conservative approach possible to find the true origin of a read is to only report those reads that align to a genome exactly once, and discard those that align multiple times equally well, for example to repeat regions (settings STAR: '--outFilterMultimapNmax 1'; settings bowtie2: MAPQ \geq 30). Except for a few single nucleotide variants (please see the answer to these SNV below), the rDNA loci encoding S2 and S3 are identical. The same is true for the A1 and A2 rDNA loci (Fig. 1d and Supplementary Table 1). Hence, when taking a conservative approach and allowing reads to only map once, reads that align to both S2 and S3 (or to A1 and A2) would be discarded. We therefore took an approach where we masked one of the two loci (in the revised version, for all analysis that is A1 and S3, from the end of the upstream region to the beginning of the downstream region of the gene) by replacing it with a series of 'N' nucleotides. This allows for read alignments to the different rDNA loci to be retained while at the same time using the most stringent mapping parameters. To highlight this approach, we provide an example analysis of the sporozoite RNA-seq sample generated in this study below (Figure 1, left). Using our approach (a), read counts of uniquely mapping reads for the 28S rRNA are only provided for S1, S2 and A2. S3 and A1 are masked and therefore not reported. Alternatively (b), only reporting uniquely mapped reads in an unmasked genome leads to substantial drop of read counts at the A1/A2, S2/3, as most reads can map equally well to both loci and are therefore not reported, therefore leading to an underestimation of A1/2 and S2/3 counts. As a third alternative (c), allowing reads to map multiple times leads to alignments of the same read being reported for the A1 and A2 (S2 and S3) loci, effectively doubling the total read count estimation. The over- and underestimation of read counts for the two-copy A1/2 and S2/3 loci in approach b) and c) are particularly obvious and exacerbated when compared to a single locus gene (i.e. S1, Figure 1, right), whose read counts do not differ depending on the mapping approach used, but whose relative contribution to the total rRNA estimate varies substantially.

The same effects on mapping approaches are also true for ChIP-seq experiments, where allowing reads to map to multiple regions introduces substantial biases in enrichment calculation. We have substantially reviewed these approaches and biases previously⁵, which have similarly been reported also for mammalian cells.⁶

Altogether, it is true that our approach loses the positional information of whether a read truly originates from either the S2 or S3 (or A1/A2) loci. This is simply a technical challenge of

studying these loci. However, at the same time, our approach maintains the exact quantitative information needed for precise RNA expression and ChIP enrichment calculations. We do not believe that this is a drawback of our study or any study of genomic loci with high sequence similarity. On the contrary, given the increasing interest in the regulation of rRNA expression and ribosome heterogeneity in *Plasmodium*^{1,2,7-10}, this study provides a comprehensive pipeline that allows for the stringent analysis of next-generation sequencing data related to rRNA expression.

> In Fig S5A and Line179-182. The author compared the transcript level of Nico gene in different stages. Compared to the level of Nico transcript in the asexual stage, the level of Nico transcript in the oocyst decreases, but the level of Nico transcript in the sporozoite increases. Given the high expression (de-repression) of S-type rRNA in both the oocyst and sporozoite at the mosquito stage, I am not convinced by the explanation that Nico-mediated NAD⁺ metabolism regulates the de-repression of S- type rRNA.

We agree with the reviewer that Nico is unlikely the primarily responsible regulator of NAD⁺ levels during host to vector transmission, as we had already stated (now on lines **213ff**). This is why we additionally investigated LDH1, which is the central enzyme responsible for aerobic glycolysis and the main driver of NAD⁺ regeneration in asexually replicating parasites. In the context of our study, we had investigated Nico because it is one of the most down-regulated genes when parasites are cultured at 32°C. Given that the knockout of Nico does lead to an upregulation of silent rRNA in asexual parasites, we concluded that Nico-mediated NAD⁺ metabolism is partially responsible for the rRNA upregulation in asexual parasites (not mosquito stages) and provided the initial lead to investigate NAD⁺ metabolism in greater detail. Yet, Nico

itself is unlikely the responsible driver across the lifecycle, exactly because it doesn't show an expression profile in the mosquito that matches the observed phenotype.

> Line 101-102 and Fig S3C. What are the A695T and C3558T? No information provided in the legend. It is confused ! In addition, why are there two SNPs from the same A1-type rRNA locus of the cloned parasite line?

As described above, we had masked the A1 and S3 locus in the genome to provide an exact absolute and relative count of rRNA expression. The 28S sequences of A1 and A2, as well as S3 and S2, differ by only a few single nucleotide variants (SNV). Masking these loci means that reads that truly originate from the S3 locus will map to S2 and reads that truly originate from A1 will map to A2, also in regions where there is a single nucleotide difference. We counted the coverage of these SNVs, which gives an estimate of the relative contribution of rRNA that is in fact transcribed from the A1 or A2 locus, and then checked whether this ratio changes across the lifecycle compared to the asexual sample. We find that in oocysts and sporozoites, more rRNA is transcribed from the A1 locus than in asexual and gametocytes. This means that those are not SNPs from different cell lines, but from two different loci of a two-copy gene within the same genome. We have added text to figure legend of Supplementary Fig. 1b and extended the Material and Methods on line **1087ff** to make this more clear to the future reader.

> Line 108-110. While the H3K9ac analysis results are shown for A2 locus in Fig1E, but not for A1 locus. Similarly, while the H3K9ac analysis results are shown for S1 and S3 locus in Figure S1C, but not for S2 locus. Throughout the manuscript, the author calls the A-type and S-type, it should be included for each gene locus.

Please see our comments on the use of A1/2 and S2/3 and data analysis above.

> In Line132, Figure S1D should be Figure S1E?

Thank you, we corrected this in the revised version.

> Figure S1E, it is necessary to highlight the position of different rRNA loci in the chromosome. It is good for the reader to check the interaction between different loci. In the current figure, it is hard to check.

We have added the respective rDNA location similar to Fig. 1g.

> Line 137-139 : What is the subcellular localization of HMGB1 and HMGB2 in the asexual blood stage? Since the author generated the protein-tagged parasite line, it is feasible to obtain these data, which could strengthen the conclusion that these two proteins act as nuclear/nucleolar proteins in gene expression regulation.

We have performed Immunofluorescence analysis (IFA) that shows the nuclear localization of HMGB1 (new Supplementary Fig. 2f) and on line 154ff. These data strengthen our conclusion that these proteins are nuclear.

> Figure S2C : explain the difference between the expected size (1146 bp) and detected size (700 bp) of PCR product

We might have loaded too much PCR product to the gel, leading to the smiling feature of the band. When reading a DNA gel, one must look at the upper-most edge of a band, which in this case runs closer to the expected size. This band is completely absent from the WT control. However, the uncropped image in Figure S8 also shows that we could have run the DNA gel a bit longer to get a better resolution. Either way, the Western Blot in Figure S2E shows that the protein runs at the expected molecular weight.

> Line 147-152. The transmission of the parasite from the human host to the mosquito vector is from 37°C to 22-25°C, the latter is commonly used for malaria-infected mosquitoes in the lab facility. The author cultured the asexual blood parasite at 32°C. This mimic is far from the physiological condition for the parasite transmission.

We have chosen to perform the experiments at 32°C since we observed that this is the lowest temperature the parasite is still able to complete the asexual replicative cycle (Supplementary Fig. 2h). This is important, since it allowed us to synchronize these parasites and avoid confounding factors that can arise from arresting cells.

As we now describe on lines 162ff and 360ff, and in the response to the reviewer above, decreasing the temperature for asexually replicating parasites is an experimental *model* we chose to upregulate S-type rRNA as described previously (¹⁻⁴), to then decipher the underlying

molecular mechanisms. Since mosquitoes experience a wide range of temperatures during the day and night, so do the parasites that develop inside them.

In fact, the results of our study show that rRNA regulation is not temperature-regulated, but metabolism-dependent. As we now describe on line **215ff**, first evidence for this was also provided by Fang et al.⁴ who found that silent rRNA can be upregulated under low glucose conditions as well as upon Sir2a disruption³. Our data indicate that a decrease in temperature for asexually parasites only leads to a disruption of their NAD⁺ metabolism and thus the chromatin landscape that is shaped by Sir2a, a process that naturally happens when the parasite switches from aerobic glycolysis to mitochondrial respiration after host-to-vector transmission. RNA-seq data across early mosquito development further show that S-type rRNA are not upregulated immediately after transmission and the accompanying temperature drop^{11,12}, further indicating that temperature is not the underlying factor regulating rRNA transcription, but that it is metabolism-dependent. We now further emphasize this point on line **425ff** of the revised manuscript. We discuss our working model on lines **399ff** of the manuscript and now also added a working model (Fig. 3i) to make it even clearer to the future reader. We further added to the discussion regarding the effect of temperature on rRNA homeostasis and the degree of rRNA levels on lines **362-403** of the revised manuscript.

> L154-155 and Fig S3C. Different contents were described in the text and figure, it is not match.

Now Supplementary Fig. 3d shows whether a temperature decrease changes the amount of rRNA that is transcribed either from the A1 or A2 rDNA locus based on the sequencing coverage of the two SNV (see above).

> L156-157 : The HMGB1 enrichment at rRNA locus at 32°C over 37°C was shown only for S3 locus, but not for S1 and S2 loci. Please provide the results for S1 and S2 loci.

Please see the logic behind the combined analysis of S2/3 above. We added the respective panel representing the S1 locus as new Supplementary Fig. 3b and describe the results in line **173ff**.

> Figure S3l : Explain the difference between the expected size and detected size of PCR product.

As with the reviewer's similar concern above, we might have loaded too much DNA to the gel, which led to a smiling feature of the band and makes it appear at a smaller size. This band is also unique to the Nico cell line and absent from the WT. In addition, the Nico protein runs at the expected molecular weight.

> Figure S2D, Figure S2E, Figure S3K, and Figure S6I. The protein marker showing the size of Aldolase is different in different blots!

As can be seen in Figure S8 of the uncropped plots, Aldolase runs at the correct size, yet we mislabeled the plots. Thank you for catching this, we corrected it in the revised version.

> Line 208-209 and Figure S4B. The images of the Giemsa-stained parasite were cited in the text, but do not support the conclusion (de-repression of rRNA) in the text.

We have added the Giemsa images to show the developmental stage of the parasites when they were harvested for the respective experiment (here: LDH1 knockdown).

> Line 217, NAD⁺ is an essential cofactor for Sirtuins, not for the HDACs.

We clarified the language to avoid any confusion for the future reader. Please also see our reply below.

> Line 215, 290, 354. Sirtuins and HDAC are different kind of deacetylase. It is not right to use the "HDAC Sir2a".

Throughout the literature, sirtuins are categorized as class III, NAD⁺ dependent histone deacetylases ('HDACs'). For a recent review see for example ref. ¹³

> In Figure S6C. Sir2a is expressed as a nuclear protein in the asexual blood stage, but the author failed to detect the Sir2a-HA in the nuclear fraction. The author generated the Sir2a-madID parasite line, but did not provide any evidence showing the nuclear localization of the Sir2a-madID protein.

We have attempted to perform IFA of both the Sir2a-HA and Sir2a-madID-HA cell lines, but we have unfortunately failed to detect any signal at all. Given that our protocol using this antibody worked for the HMGB cell lines (see above), this might be a Sir2a-specific issue. It is also possible that Sir2 proteins are expressed at low levels, making detection with antibody-specific methods difficult. However, given the significant results from the histone mass-spectrometry results (Fig. 2j,k) and the specific signal of the Sir2a-madID experiments (compared to the nuclear-targeted madID control [Supplementary Fig. 6i]) throughout Fig. 3 and Supplementary Fig. 6j, we provide multiple orthogonal evidence for the nuclear localization of Sir2a(-madID) in either cell line. In addition, the (peri-)nuclear localization and mode of action has already been shown in previous studies^{3,14-16}.

> Line 277-279. In direct comparison to the active rDNA locus, however, Sir2a occupancies are 2- to 20-fold higher upstream of all major silent rDNA loci (Figure 3H, S7G). How did the author calculate the 2- to 20-fold? Have no idea from Figure 3H and S7G.

We are taking the AT-content adjusted frequency of detected 6mA sites over a window of 100bp (line **1356ff**). The fold change was calculated based on the maximum 6mA frequency in the upstream region of rDNA loci and the bedgraph file provided at GEO (accession: GSM8825930). To make it more accessible, we have added these values to the new Supplementary Table 11. We have now also included a table representing regions that feature a ≥ 10 -fold enrichment in 6mA between the Sir2a-madID and madID control samples (Supplementary Table 9). The calculation of these fold-changes is described on line **1375ff** of the revised manuscript.

> L341. Figure S7F is a schematic of the working model, but it was cited here. Wrong?

New Supplementary Fig. 7e (original Figure S7F) is not a figure of our working model, but the workflow of the quantitative ChIP-seq (ChIP-Rx) showing how adding equal amounts of spike-in chromatin to a sample of interest allows for the absolute quantification of peak enrichment between samples and replicates (see lines **312ff**). We have extended the explanation of this Figure in the figure legend and the methods section on line **1208ff**. We now also include a working model of our findings in Fig. 3i.

> Many errors in the figure and text.

After careful reading by all authors and two native English speakers, we hope to have found all mistakes that were present in the original version and corrected them in the revised manuscript.

Reviewer #2 (Remarks to the Author):

The submission from Couble and colleagues describes the molecular basis underlying the transition between alternative ribosomal RNA expression during differentiation by the human malaria parasite *Plasmodium falciparum*. Malaria parasites that infect a range of hosts all display a unique property of having rRNAs encoded by a handful of genomic loci rather than the large copy number arrays that are typical of eukaryotic organisms. Even more unusual, these organisms activate different rRNA genes when transitioning between their mammalian and mosquito hosts. The biological significance underlying these rRNA transitions is not known, however changes in glucose availability and temperature were previously shown to be important cues leading to changes in rRNA expression. Here the authors identify shifts in metabolism resulting in changes in NAD⁺ availability as a key contributor to this change in rRNA expression, and further show the importance of altered activity of the histone deacetylase Sir2a.

The authors provide a substantial amount of data to support their model and overall, the study is an important contribution to our understanding of this fascinating phenomenon. However, the manuscript would benefit from some additional clarity in several parts to further reinforce the authors' conclusions and to not overstate the effect that they document. Specific comments are below:

1. Much of the manuscript is focused on understanding the activation of expression of the S2/3 rDNA genes through changes in temperature or metabolic manipulations. Throughout the manuscript, this is shown as Fold Change in RNA. Several previously published studies have documented this phenomenon and shown changes in S2/S3 expression up to 30-50-fold (for example Fang et al, JBC 2004 and Samuel et al, Pathogens, 2024). Here the authors shown a consistent, although smaller change in expression. For example, Figure 2A displays an increase in S2/S3 expression of around 16-fold. Do the authors think the lower change here is because they limited the temperature drop to 32 degrees rather than the 27 degrees used in previous studies?

We had chosen to perform our experiments at 32°C because we found that this is the lowest temperature the parasite is still able to complete the asexual replicative cycle. This was important for us to be able to synchronize the parasite and avoid possibly confounding factors due to cell cycle arrest, which we found usually occurs in the early trophozoite stage (~20 hours post infection). It is conceivable that lower temperatures lead to an even greater perturbation of the cell's metabolism (and possibly be the reason for the observed cell cycle arrest) and therefore stronger upregulation of S-type rRNA. However, even a 50-fold increase observed previously at lower temperatures (¹⁻⁴) still falls far short of the S-type rRNA levels measured in the mosquito.

Of note, our goal and experimental setup was not to resolve ribosome heterogeneity (and we have also adjusted our title accordingly), but to define the states of active and silent rDNA chromatin and use different perturbations to measure the chromatin dynamics underlying the consistent pattern of rRNA expression we found throughout the study. We now discuss this point in greater detail on line **392ff** and discuss the differences of our approach and previous studies on line **360ff**. Please see also our reply directly below in this regard.

Similarly, the manipulations of LDH levels, Sir2a levels or Nicotinamidase levels resulted in changes in expression by only 3-4-fold, much less than that observed previously. The authors note in the Discussion that “S-type rRNA do not reach the same absolute expression levels as they normally do in mosquito stages” and note the possible contribution of additional activation of the TCA cycle in mosquito stages (e.g. through α -ketoglutarate). However, it is not clear why the manipulations of Sir2a/LDH/Nicotinamidase do not meet the level of change shown in Figure 2A. Is it possible that these conditions are important for keeping the loci silent but are not sufficient to induce a fully active state, suggesting that an additional, temperature responsive activator is missing in these experiments? Any thoughts or clarifications would be useful.

The reviewer points out an important point that we have not clarified well enough in our discussion of the manuscript.

We describe the differences of S-type transcription levels between 1) the experimental model in asexual parasites and mosquitoes (line **367ff**), 2) previous studies and ours (line **360ff**), and 3) the additive effect temperature can have on rRNA homeostasis (line **384ff**) and what our model allowed us to do and not do (line **392ff**).

As stated in the original version of the manuscript, one main reason we probably aren't seeing the same increase in upregulation in any condition (including 32°C) compared to mosquito

stages is because the presence of an activating modification (i.e. histone acetylation) might be insufficient if the repressor (i.e. HP1) is not removed. Furthermore, mosquito stage-specific ribosome biogenesis factors and auxiliary ribosomal proteins that help process and stabilize the de-repressed S-type rRNA might be lacking (**line 370ff**).

As for the comparison between the individual KDs and the 32°C treatment, one aspect might be that at 32°C, both Nico and LDH1 are downregulated, possibly compounding the effect and perturbation of NAD⁺ metabolism that can be even stronger when parasites are grown at 26°C, yet at the cost of completing the asexual replicative cycle (**line 360ff**).

It is also possible that temperature itself affects S-type rRNA levels. We have now added an important observation to line **384ff** of the manuscript: S-type ribosomes feature a substantially lower GC content than A-type ribosomes. In prokaryotes, it is known that the GC content of structural RNAs positively correlates with the optimal growth temperature. Hence, it is possible that S-type rRNA transcribed at 37°C (in KD conditions) or even 32°C (compared to 26°C used previously) might not be stably folded into mature ribosomes and is thus more quickly degraded. Thus S-type rRNA at 37°C (or even 32°C) might 1) not fold correctly or as well as at 26°C and then 2) degrade faster at non-permissive, elevated temperatures.

Hence, while temperature might be needed for stable S-type rRNA homeostasis, temperature alone cannot be considered the required factor for S-type rRNA activation, given that S-type rRNA are not upregulated until at least more than > 24h after transmission and the accompanying temperature drop^{11,12}. We have clarified this further on line **425ff** of the revised manuscript.

Overall, we now clarify these points by stating on line **213ff**, **249ff** and **399ff** of the revised manuscript that our study provides evidence for a remodeling of the rDNA chromatin environment that is dependent on the metabolic state of the cell, making silent rDNA loci permissible for transcription. Yet the full transcriptional activity on these loci and the S-type ribosome maturation might depend on additional, stage-specific factors that could not be fully resolved in our experimental setup (**360-403**). We also reflect these findings in our title, using the term 'promote' rather than 'regulate' or 'control'.

2. The authors described the chromatin landscape at active and silent rDNA loci. This is a primary aspect of the paper. However, the data presented are not complete. For example, H3K9ac is noted as an important mark for transcriptionally active chromatin and a peak is shown upstream of the active A2 locus (Figure 1E), however no data are presented for the A1 locus.

As this is a concern that was also raised by reviewer 1, we will explain our reasoning for ‘merging’ A and S type loci in detail here and have also expanded and clarified our approach on line **1045ff** of the revised manuscript.

Every distinct molecule sequenced in an Illumina run can only originate from one specific position of the physical genome. Short read aligners such as those used here (STAR for RNA-seq, bowtie2 for ChIP-seq) attempt to find and report this specific position with the highest confidence possible (i.e. the MAPQ score reported as $Q = -10 \cdot \log_{10}(p)$, where p equals the probability that the reported alignment is not the true location from where the sequenced molecule originated). Thus, the most conservative approach possible to find the true origin of a read is to only report those reads that align to a genome exactly once, and discard those that align multiple times equally well, for example to repeat regions (settings STAR: ‘--outFilterMultimapNmax 1’; settings bowtie2: MAPQ \geq 30). Except for a few single nucleotide variants (please see the answer to these SNV below), the rDNA loci encoding S2 and S3 are identical. The same is true for the A1 and A2 rDNA loci (Fig. 1d and Supplementary Table 1).

Hence, when taking a conservative approach and allowing reads to only map once, reads that align to both S2 and S3 (or to A1 and A2) would be discarded. We therefore took an approach where we masked one of the two loci (in the revised version, for all analysis that is A1 and S3, from the end of the upstream region to the beginning of the downstream region of the gene) by replacing it with a series of ‘N’ nucleotides. This allows for read alignments to the different rDNA loci to be retained while at the same time using the most stringent mapping parameters. To highlight this approach, we provide an example analysis of the sporozoite RNA-seq sample generated in this study below (Figure 1, left). Using our approach (a), read counts of uniquely mapping reads for the 28S rRNA are only provided for S1, S2 and A2. S3 and A1 are masked and therefore not reported. Alternatively (b), only reporting uniquely mapped reads in an un-masked genome leads to substantial drop of read counts at the A1/A2, S2/3, as most reads can map equally well to both loci and are therefore not reported, therefore leading to an underestimation of A1/2 and S2/3 counts. As a third alternative (c), allowing reads to map multiple times leads to alignments of the same read being reported for the A1 **and** A2 (S2 **and** S3) loci, effectively doubling the total read count estimation. The over- and underestimation of read counts for the two-copy A1/2 and S2/3 loci in approach b) and c) are particularly obvious and exacerbated when compared to a single locus gene (i.e. S1, Figure 1, right), whose read counts do not differ depending on the mapping approach used, but whose relative contribution to the total rRNA estimate varies substantially.

The same effects on mapping approaches are also true for CHIP-seq experiments, where allowing reads to map to multiple regions introduces substantial biases in enrichment calculation. We have substantially reviewed these approaches and biases previously⁵, which have similarly been reported also for mammalian cells.⁶

Altogether, it is true that our approach loses the positional information of whether a read truly originates from either the S2 or S3 (or A1/A2) loci. This is simply a technical challenge of studying these loci. However, at the same time, our approach maintains the exact quantitative information needed for precise RNA expression and CHIP enrichment calculations. We do not believe that this is a drawback of our study or any study of genomic loci with high sequence similarity. On the contrary, given the increasing interest in the regulation of rRNA expression and ribosome heterogeneity in *Plasmodium*^{1,2,7-10}, this study provides a comprehensive pipeline that allows for the stringent analysis of next-generation sequencing data related to rRNA expression.

Similarly, the absence of this mark is shown in supplemental Figure S1C for the S1, S3 and O2 loci, but S2 and O1 are not shown. This is despite the text stating that a peak of H3K9ac is not found “at any of the silent rDNA loci in asexually replicating cells.”

Please see our response above. We have added the O1 locus to Supplementary Fig. 1c,d.

Similarly, the distribution of HP1 and ATAC-seq accessibility obtained from previously published datasets omits the A1 and S2 loci. Is there a reason these data are not available or not included in the analysis shown in the paper? As currently written, the analysis appears incomplete.

For both HP1 and ATAC seq, we have re-analyzed the initial data by mapping them to the masked genome explained above; thus, the S3 and A1 loci are omitted from the analysis. These graphs are now shown for all other loci in Fig. 1e,f and Supplementary Fig. 1c,d. We have further extended the analysis of the ATAC-seq data on line 112ff and 326ff, and describe the analysis on line 1229f.

3. The authors note that in response to a shift in temperature to 32 degrees, HMGB1 is recruited to the S3 locus (Figure 2B). This figure does indeed show a somewhat subtle, but clear increase in HMGB1 at 32 degrees. They then show very little association of HMGB1/2 at the silent loci in Figure S2G. Presumably the data in Figure S2G are from parasites grown at 37 degrees, although the Figure legend does not make this clear.

Yes, this is correct. We have clarified our Figure legends in this regard in the new Supplementary Fig. 2g which shows the HMGB1/2 coverage for the S1, S2, O1 and O2 loci at 37°C.

More importantly, does the presence of HMGB1 or 2 increase at these loci when the temperature is decreased to 32 degrees? The text says mentions recruitment of HMGB1 to the S2/S3 loci at 32 degrees and cites both Figures 2B (which shows the S3 locus) and Figure S2F (which I think should be S2G, and which only shows the absence of HMGB1 and 2 at four loci). Thus, the data are somewhat incomplete and do not directly coincide with what is stated in the text.

As stated above, due to the usage of a masked genome, the S3 locus is omitted from the analysis. However, we now show the changes of HMGB1 coverage between 37°C and 32°C for the S2 and S1 loci in Fig 2b and Supplementary Fig. 3b, respectively. The quantification of these graphs as measured by macs2 bdgdiff are also shown in Supplementary Table 5.

Similarly, the noncoding “truRNAs” upstream of the S3 locus (Figure 2B) have also been reported to be upregulated in response to a temperature drop (Sharma et al, Genes, 2023 and Samuel, Pathogens, 2024), however there appears to be no change in HMGB1 occupancy. Thus, the conclusions are somewhat muddled by the lack of complete data or a bit inconsistent

with the stated model. A more complete explanation accommodating all the available data would be helpful.

In our previous analysis we have used a reference genome that has masked only the actual rDNA locus, i.e. from the 5' 18S to the 3' of the 28S. We have now performed all analysis again, using a genome version that also masked the upstream regions of the A1 and S3 loci, i.e. from the 3' end of the upstream gene to the 5' end of the downstream gene. This allowed us to now also stringently map reads to the highly similar upstream regions of S2/3 (and thus also providing evidence for the efficiency of our approach). We are now able to clearly detect an increase in HMGB1 coverage at these upstream regions encoding for the truRNAs. Similar to previous reports, we also find that the upstream ncRNA are upregulated when temperatures are decreased (new Supplementary Fig. 3c). To complete the picture of ncRNA transcription, we have now also performed full-length total RNA sequencing from parasites cultured at 32°C. These data (now included with Fig. 2b) show that the truRNA are transcribed as one long, contiguous transcript that is clearly separated from the S2 pre-rRNA. We now also report these data together with the quantitative ChIP-seq results in Fig. 3h and on line 177ff and 326ff.

4. The authors knocked down Nicotinamidase expression and noted a de-repression of previously silent S-type rRNA, which on line 172 they note is “similar to that which takes place upon growth at 32 degrees.” However, as mentioned in point 1 above, it should be noted by the authors here that the change observed is a fraction of what was observed at 32 degrees, a little over 3-fold as compared to the ~16-fold observed by the temperature shift. A similar magnitude effect is observed in the LDH1 and Sir2a knockdowns.

The reviewer is correct, and we apologize for not having used language that is precise enough. We observe a similar pattern of rRNA transcription, i.e. S3 > S1 and no change in A-type. We have corrected this on line 362ff. Please also see our response above regarding why we might not observe a similar increase in rRNA levels in the KD conditions compared to the low temperature conditions.

5. The authors mapped the interactions of Sir2a throughout the genome and noted that it appears to be preferentially associated with euchromatic regions of the genome, suggesting a possible role in modulating histone acetylation in transcriptionally permissive chromatin regions (lines 251-158). They then note the “exception” of var introns and mention var intron-derived

transcripts. The logic here is not clear. If transcripts come from var-introns (and thus they are transcriptionally permissive), then the association of Sir2a with var introns would seem to be consistent with its association with euchromatic (and transcriptionally permissive) regions elsewhere in the genome. It is not clear why the authors consider this an “exception”.

The reviewer is correct, and we were not precise enough on this point. On a genome-wide perspective, var genes (and thus var introns) are located within heterochromatic regions, whereas (again, on a genome-wide scale), we find Sir2a mainly in euchromatic regions. As the var-intron shows, however, being located within a heterochromatic region does not mean that a specific locus might not be transcriptionally permissive. We have clarified our language in this regard on line **296ff** of the revised manuscript.

Reviewer #3 (Remarks to the Author):

With the presented article “A metabolism-chromatin axis promotes ribosome heterogeneity in the human malaria parasite” the authors provide an intriguing and timely manuscript. Using a combination of forward genetics, metabolomics and state-of-the-art chromatin/epigenetics tools and experiments, the authors provide compelling mechanistic insights how a link between the metabolic state of the parasite and the epigenetic machinery controlling gene silencing ensure the stage specific expression of divergent ribosomes in *Plasmodium falciparum*.

The authors achieve this by providing insights on how the epigenetic state/transcriptional repression of the different rRNA genes is influenced by culturing the parasites at lower temperature and then link these changes to the NAD⁺/NAM levels in the parasite using targeted metabolomics and conditional KO parasites for the nicotinamidase NICO, a key enzyme in the NAD⁺ metabolism. Using an additional conditional KO line for the LDH1 protein, the authors are further able to recapitulate the decreased temperature phenotype / NICO knockdown phenotype, indicating how the shift from aerobic glycolysis to more oxidative phosphorylation during the build-up to the human to mosquito transmission could be the driving force behind this metabolic control of rRNA expression. Finally, the authors can present the histone deacetylase Sir2a as likely direct target of the changing NAD⁺/NAM ratios and key player in “translating” these changes in NAD⁺ levels into transcriptional regulation of the rRNA genes by changes in chromatin modifications, namely histone acetylation. While the presented effect of changes in the glucose metabolism on rDNA transcription and the involvement of Sir2a as a key regulator of rDNA transcription have been investigated before (as cited by the authors), the authors do a

nice job in connecting these observations and add significant new mechanistic insights using the NICO, LDH1 and Sir2a KO parasites lines. The methodology is sound; all experiments have been conducted with the appropriate number of biological replicates and controls. In addition, the used methods of Micro-C and Sir2a-madID sequencing go beyond the “state-of-the-art” in the field and nicely showcase the adaption of these methods for molecular research in *P. falciparum*.

In summary, the presented manuscript will be highly significant to the field and nicely complements recent insights in the regulatory link between metabolism and the epigenetic regulation of sexual development and immune evasion in *P. falciparum* (see e.g. Brancucci NMB et al., 2017, Schneider VM et al., 2023) At the same time, the link between gene regulatory/epigenetic processes and metabolism/metabolic shifts are current topics in model organisms as well as medically relevant fields such as cancer epigenetics. The presentation of possible common principles or markedly different adaptations governing these processes from model organisms to diseases phenotypes and pathogenic single celled organisms like *P. falciparum*, will thus also be of significant interest to a broader audience.

Minor comments:

In general, I would appreciate, if the chromosome numbers would be added in addition to the genomic coordinates in the ChIP / madIDseq graphs. This would make orientation in the dataset a bit easier.

We have added the chromosome numbers to all genome-wide graphs throughout the manuscript.

Line 135 onwards:

“To identify proteins potentially involved in the structuring of the rDNA chromatin environment and/or recruitment of PolII to active rDNA loci, we annotated the PolII machinery in *P. falciparum*. This search identified high-mobility group proteins 1 (HMGB1 PF3D7_1202900) and 2 (HMGB2, PF3D7_0817900) as putative homologs of human nucleolar transcription factor 1 (Table S2). This protein is required for PolII recruitment and subsequent remodeling of active rDNA

chromatin and rRNA transcription²³. Epitope-tagging of the two proteins with 3x hemagglutinin (3xHA) (Figure S2A-E) followed by ChIP-seq revealed that they specifically associate with the active, but not the silent rDNA loci (Figure 1H,I Figure S2F,G, Table S3), providing evidence for an orthogonal role of these proteins similar to other eukaryotic nucleolar transcription factors.”

I would not define HMGB1/2 as homologs to UBF1. The presented blast hit is due to the presence of the HMG-box motif in UBF1. However, UBF1 is a bigger multidomain protein and not comparable to HMGB1/2 in Pf. A better comparison here might be the described role of yeast HMO1 in rDNA regulation (Huffines AK, 2024). I would thus suggest changing the wording in this paragraph.

We thank the reviewer for making us aware of this work and HMO1. We have included HMO1 and other proteins annotated with GO:0006356 (regulation of transcription by RNA polymerase I) in a new blast database. In this new search, the reciprocal best hit of *Plasmodium* HMGB1 and 2 is still human UBF1 (Supplementary Table 2). However, we have rephrased the main text in our revised manuscript and include the study HMO1 and rDNA regulation on line **141ff**.

A recent study characterized the HMGB1 protein in asexual stage of P.f. (Lu B, et al., Parasitology, 2021). In the study the authors generated a HMGB1 KO parasite line which showed no apparent growth defect or any remarkable changes in rDNA expression regulation (as far as this reviewer is aware). Do the authors have an idea how this fits with the proposed role of HMGB1 as nucleolar transcription factor?

HMGB1 and HMGB2 are highly similar on the protein level (57% of all positions are identical, see below). Together with our finding that both proteins locate to the same genomic region in asexual parasites (i.e. the A-type rDNA locus), it is possible that these proteins perform overlapping functions and that a single gene knock-out can be compensated by the other protein.

```
CLUSTAL 2.1 multiple sequence alignment

PF3D7_0817900.1-p1    MASKSQKKVLKQNKKKKKDPLAPKRALSAYMFYVKDKRLEIIEKEPELAKDVAQVGKLI
PF3D7_1202900.1-p1    MKN-TGKEVRKR--RKNKDPHAPKRSL SAYMFFAKEKRAEIIISKPELSKDVATVGKMI
* . : *!* *! :!*!!!! *!!!!*!!!!*!* *!* *!*!!!!*!!!! *!!!!*

PF3D7_0817900.1-p1    GEAWGQLSPAQKAPYEKKAQLDKVRYSEIEEYRKKNQE--
PF3D7_1202900.1-p1    GEAWNKLGEKEKAPFEKKAQEDKLRYEKEKAEYANMKMKA-
*!*.!* . :!*!!!! *!!!!* *! *! *! : : :
```

Fig S7A:

Was the var intron peak the only discernible enrichment of Sir2A in the genome? I couldn't find a table or similar with possible Sir2A peaks. In case there were additional peaks detectable such a table overview would be interesting and could be added.

We have now added a table showing all regions featuring a fold-change enrichment ≥ 10 between the Sir2a-madID and madID control with their annotation to Supplementary Table 9 of the manuscript. A description of how this enrichment was calculated can be found on line 1359ff of the revised manuscript.

In summary I enjoyed reviewing the presented manuscript, congratulate the authors to a great study and would recommend publication with these minor revisions.

Best regards.

REFERENCES

1. Sharma, I., Fang, J., Lewallen, E.A., Deitsch, K.W., and McCutchan, T.F. (2023). Identification of a long noncoding RNA required for temperature induced expression of stage-specific rRNA in malaria parasites. *Gene* 877, 147516. <https://doi.org/10.1016/j.gene.2023.147516>.
2. Samuel, H., Campelo Morillo, R., and Kaf sack, B.F.C. (2024). Suppression by RNA Polymerase I Inhibitors Varies Greatly Between Distinct RNA Polymerase I Transcribed Genes in Malaria Parasites. *Pathogens* 13, 924. <https://doi.org/10.3390/pathogens13110924>.
3. Mancio-Silva, L., Lopez-Rubio, J.J., Claes, A., and Scherf, A. (2013). Sir2a regulates rDNA transcription and multiplication rate in the human malaria parasite *Plasmodium falciparum*. *Nat Commun* 4, 1530. <https://doi.org/10.1038/ncomms2539>.
4. Fang, J., Sullivan, M., and McCutchan, T.F. (2004). The effects of glucose concentration on the reciprocal regulation of rRNA promoters in *Plasmodium falciparum*. *J Biol Chem* 279, 720–725. <https://doi.org/10.1074/jbc.M308284200>.
5. Baumgarten, S., and Bryant, J. (2022). Chromatin structure can introduce systematic biases in genome-wide analyses of *Plasmodium falciparum*. *Open research Europe* 2, 75. <https://doi.org/10.12688/openreseurope.14836.2>.
6. Marinov, G.K., Wang, J., Handler, D., Wold, B.J., Weng, Z., Hannon, G.J., Aravin, A.A., Zamore, P.D., Brennecke, J., and Toth, K.F. (2015). Pitfalls of mapping high-throughput

- sequencing data to repetitive sequences: Piwi's genomic targets still not identified. *Dev Cell* 32, 765–771. <https://doi.org/10.1016/j.devcel.2015.01.013>.
7. McGee, J.P., Armache, J.-P., and Lindner, S.E. (2023). Ribosome heterogeneity and specialization of Plasmodium parasites. *PLoS Pathog* 19, e1011267. <https://doi.org/10.1371/journal.ppat.1011267>.
 8. Erath, J., Djuranovic, S., and Djuranovic, S.P. (2019). Adaptation of Translational Machinery in Malaria Parasites to Accommodate Translation of Poly-Adenosine Stretches Throughout Its Life Cycle. *Front Microbiol* 10. <https://doi.org/10.3389/fmicb.2019.02823>.
 9. Erath, J., Kemper, D., Mugo, E., Jacoby, A., Valenzuela, E., Jungers, C.F., Beatty, W.L., Hashem, Y., Jovanovic, M., Djuranovic, S., et al. (2025). A rapid, simple, and economical method for the isolation of ribosomes and translational machinery for structural and functional studies. *Nat Commun* 16, 7185. <https://doi.org/10.1038/s41467-025-62314-8>.
 10. Pavlovic Djuranovic, S., Erath, J., Andrews, R.J., Bayguinov, P.O., Chung, J.J., Chalker, D.L., Fitzpatrick, J.A., Moss, W.N., Szczesny, P., and Djuranovic, S. (2020). Plasmodium falciparum translational machinery condones polyadenosine repeats. *Elife* 9. <https://doi.org/10.7554/eLife.57799>.
 11. Mohammed, M., Dziejniech, A., Sekar, V., Ernest, M., Alves E Silva, T.L., Balan, B., Emami, S.N., Biryukova, I., Friedländer, M.R., Jex, A., et al. (2023). Single-Cell Transcriptomics To Define Plasmodium falciparum Stage Transition in the Mosquito Midgut. *Microbiol Spectr* 11, e0367122. <https://doi.org/10.1128/spectrum.03671-22>.
 12. Yan, Y., Cheung, E., Verzier, L.H., Appetecchia, F., March, S., Craven, A.R., Du, E., Probst, A.S., Rinvee, T.A., de Vries, L.E., et al. (2024). Mapping Plasmodium transitions and interactions in the Anopheles female. *bioRxiv*. <https://doi.org/10.1101/2024.11.12.623125>.
 13. Park, S.-Y., and Kim, J.-S. (2020). A short guide to histone deacetylases including recent progress on class II enzymes. *Exp Mol Med* 52, 204–212. <https://doi.org/10.1038/s12276-020-0382-4>.
 14. Merrick, C.J., and Duraisingh, M.T. (2007). Plasmodium falciparum Sir2: an unusual sirtuin with dual histone deacetylase and ADP-ribosyltransferase activity. *Eukaryot Cell* 6, 2081–2091. <https://doi.org/10.1128/EC.00114-07>.
 15. Duraisingh, M.T., Voss, T.S., Marty, A.J., Duffy, M.F., Good, R.T., Thompson, J.K., Freitas, L.H., Scherf, A., Crabb, B.S., and Cowman, A.F. (2005). Heterochromatin silencing and locus repositioning linked to regulation of virulence genes in Plasmodium falciparum. *Cell* 121, 13–24. <https://doi.org/10.1016/j.cell.2005.01.036>.
 16. Freitas-Junior, L.H., Hernandez-Rivas, R., Ralph, S.A., Montiel-Condado, D., Ruvalcaba-Salazar, O.K., Rojas-Meza, A.P., Mâncio-Silva, L., Leal-Silvestre, R.J., Gontijo, A.M., Shorte, S., et al. (2005). Telomeric Heterochromatin Propagation and Histone Acetylation Control Mutually Exclusive Expression of Antigenic Variation Genes in Malaria Parasites. *Cell* 121, 25–36. <https://doi.org/10.1016/j.cell.2005.01.037>.

Reviewer #1 (Remarks to the Author):

> One major concern remains unaddressed.

Throughout the manuscript, the author only emphasized the upregulation of S-type (S1, S2, and S3) rRNA in the asexual blood stage in temperature or metabolic manipulations, but did not fully present the extent or level of the rRNA upregulation.

In the Supplementary Table 4, the results showed that the transcript levels for the S-type (S1, S2, and S3) rRNA are 30000-50000 (in FPKM) in oocyst and sporozoite. These data show the real physiological level for S-type (S1, S2, and S3) rRNA expression. In either the introduction or discussion section of the manuscript, this information is not included. It is necessary to highlight and describe these information in the manuscript (for the full understanding of readers).

In the Supplementary Table 4, the results also showed that after either 32°C temperature or knockdown of Nico, LDH, or Sir2a, the upregulated levels (ranging from 25 to 254) of S-type rRNA are still quite lower than the level (30000-50000 in FPKM) of S-type rRNA in oocyst and sporozoite. Even after a 10-fold upregulation, it is still less than 1% of the physiological expression level (30000-50000 in FPKM) in the oocyst and sporozoite stages!

As we have written already in the previous response and also in the current version of the manuscript (line 360ff) we acknowledge that the upregulation of repressed rRNA does not reach the levels usually observed in the mosquito, but that the metabolic perturbation is sufficient to render rDNA chromatin permissive to transcription (line 384ff).

The author showed that the environmental temperature switch or metabolism manipulation in the asexual blood stage parasite has a minor effect on the upregulation expression of S-type rRNA, but this change in transcript level is far from the real physiological level in the oocyst/sporozoite stage. For this point, I was not convinced by the update in the revised manuscript.

We have already addressed these points in the previous reply and the latest version of the manuscript starting from line 360ff.

> Some cited references from the preprint have been officially published in the journal. Need updated.

All references have been updated to the most recent versions

Reviewer #2 (Remarks to the Author):

The revised submission from Couble and colleagues provides an important contribution to our understanding of the transition between alternative ribosomal RNA expression during differentiation of malaria parasites. This is a fascinating phenomenon that has puzzled researchers for decades. By linking this process to specific metabolic changes, the manuscript provides key insights into how the parasites transition to the environment of the mosquito vector.

The initial version of the paper primarily suffered from a lack of clarity regarding how sequencing data for the different ribosomal RNA types were analyzed and presented (specifically the “masking” of specific loci) and the need for greater explanation regarding the level of de-repression observed in this experimental model as compared to actual rRNA expression levels in the mosquito. The authors have suitably addressed these concerns by expanding and clarifying the text and by modifying the figures appropriately. Overall, the data are now appropriately presented and interpreted. I have only a couple of very minor suggestions for the authors.

1. On line 53-54, the authors state “....distinct rRNA genes are assembled into possibly divergent ribosomes.....” This should probably be written “wherein the products of distinct rRNA genes are assembled into possibly divergent ribosomes...” Genes are not assembled into ribosomes, rRNAs are.

2. On line 83-84, the authors note “Thus, A2 and S2 will be representative of the A1/2 and S2/3 rDNA loci, respectively.” As mentioned above, this is an important clarification. However, some readers might be a bit confused since the first Figure shows all loci separately. Perhaps this sentence could be modified to say “Thus, when mapping sequencing reads, A2 and S2 will be representative of the A1/2 and S2/3 rDNA loci, respectively.”

We have adjusted the language of the manuscript accordingly

The Plasmodium parasite evolves several divergent copies of rRNA genes, which are localized in different chromosomes. Interestingly, the A-type (A1 and A2) rRNA shows ubiquitous expression in all the development stages, including human and mosquito host stages; however, the S-type (S1, S2, and S3) rRNA is specifically expressed in oocyst and sporozoite at the mosquito host, but not in the human host. The regulation and mechanism of mosquito stage-specific expression of S-type (S1, S2, and S3) rRNA remains elusive.

The study by Couble et al. revealed that the epigenetic signatures of rRNA transcription and found that rDNA silencing relies on aerobic glycolysis in the human host. Disruption of NAD⁺ regeneration during lactate fermentation promotes rDNA de-repression and identify the sirtuin histone deacetylase Sir2a as the mediator between fluctuating NAD⁺ levels and a functional transcriptional outcome.

Major concerns:

> For the five copies of rRNA genes, the A type (A1 and A2) rRNA shows ubiquitous expression in all the developmental stages, including human and mosquito host. However, the S type (S1, S2, and S3) rRNA is specifically expressed in oocyst and sporozoite at the mosquito host, but not in the human host. The author called the no expression of S-type rRNA in human host as gene silencing or repression, and the oocyst- and sporozoite-stage specific expression of S-type rRNA as gene de-repression. During the parasite life cycle, lots of genes show stage-specific expression. It complicates the stage-specific gene expression as “gene silencing/repression and gene de-repression”.

> The author investigated the effect of the metabolism and chromatin on the gene expression for both A-type (A1 and A2) and S-type (S1, S2, and S3) rRNA in the asexual blood stage in the human host. In the manuscript, no direct evidence showed the ribosome heterogeneity. It is not appropriate for the title “A metabolism-chromatin axis promotes ribosome heterogeneity in the human malaria”.

> In Fig 2A, the author investigated the transcript expression of A-type (A1 and A2) and S-type (S1, S2, and S3) rRNA in the asexual blood stage at 37°C and 32°C, respectively, and found that the S-type rRNA showed pronounced upregulation at 32°C. I checked the data of the normalized expression value in FPKM in Table S4 (see below).

Table S4: Normalized expression values in FPKM (Fragments per kilobase of exon per one million mapped reads) of the different 28S rRNA-types

Figure 1 A,B; Figure 2A																
Name	ID	NF54- 32C_R1	NF54- 32C_R2	NF54- 32C_R3	NF54- 37C_R1	NF54- 37C_R2	NF54- 37C_R3	Gam_R 1	Gam_R 2	Gam_R 3	Ooc_R1	Ooc_R2	Ooc_R3	Spz_R1	Spz_R2	Spz_R3
S1	PF3D7_0112	237	254	256	70	65	64	59	48	64	30981	29304	30174	58722	54601	54073
A1/2	PF3D7_0726	120113	124800	118484	124438	126459	124376	112494	109346	112925	103526	106122	102090	63621	74051	72664
O1	PF3D7_0801	19	21	19	9	8	11	9	9	6	347	537	448	6	0	0
O2	PF3D7_0830	4	8	7	2	1	4	10	12	7	280	253	237	0	0	0
S2/3	PF3D7_1371	211	233	244	12	15	15	15	9	9	37870	34502	36026	40505	37325	35943

Compared to 37°C, the transcript amount of S1 rRNA increases from 65 to 254, and S2/S3 rRNA increases from 15 to 233. However, the upregulated level of S-type (S1,

S2, and S3) rRNA is quite much lower than the level (30000-50000 in FPKM) of S-type (S1, S2, and S3) rRNA in oocyst and sporozoite. Even after a 10-fold upregulation at 32°C, it is still less than 1% of the physiological expression level (30000-50000 in FPKM) in the oocyst and sporozoite stages!

I agree that the environmental temperature switch from 37 to 32°C has an effect on the expression of S-type rRNA, but it just plays a minor role. This change in transcript level is far from the real physiological change in the transcript level between the asexual blood stage and oocyst/sporozoite stage.

> Similar concern as above

In Fig 2F, only a 2-4 fold upregulation of S-type (S1, S2, and S3) rRNA after knockdown of Nico gene.

In Fig 2H, only a 2-4 fold upregulation of S-type (S1, S2, and S3) rRNA after knockdown of LDH gene.

In Fig 2I, only a 2-2.5 fold upregulation of S-type (S1, S2, and S3) rRNA after knockout of Sir2a gene.

> In addition, lots of other genes were differentially expressed in the asexual blood stage at 32°C (see Fig 2C). The gene expression regulation by lower temperature is not specific to the S-type rRNA genes.

> A1, A2, S1, S2, and S3 rRNA are five individual genes. In many figures (Fig 2A, Fig 2F, Fig 2H, Fig 2I, and several supplementary figures), A1 and A2 were merged as one group A1-2, while S2 and S3 were merged as one group S2-3. What is the reason? All the individual rRNA genes should be analyzed and presented individually.

> In Fig S5A and Line179-182. The author compared the transcript level of Nico gene in different stages. Compared to the level of Nico transcript in the asexual stage, the level of Nico transcript in the oocyst decreases, but the level of Nico transcript in the sporozoite increases. Given the high expression (de-repression) of S-type rRNA in both the oocyst and sporozoite at the mosquito stage, I am not convinced by the explanation that Nico-mediated NAD⁺ metabolism regulates the de-repression of S-type rRNA.

> Line 101-102 and Fig S3C. What are the A695T and C3558T? No information provided in the legend. It is confused! In addition, why are there two SNPs from the same A1-type rRNA locus of the cloned parasite line?

> Line 108-110. While the H3K9ac analysis results are shown for A2 locus in Fig1E, but not for A1 locus. Similarly, while the H3K9ac analysis results are shown for S1 and S3 locus in Figure S1C, but not for S2 locus. Throughout the manuscript, the author

calls the A-type and S-type, it should be included for each gene locus.

> In Line 132, Figure S1D should be Figure S1E?

> Figure S1E, it is necessary to highlight the position of different rRNA loci in the chromosome. It is good for the reader to check the interaction between different loci. In the current figure, it is hard to check.

> Line 137-139: What is the subcellular localization of HMGB1 and HMGB2 in the asexual blood stage? Since the author generated the protein-tagged parasite line, it is feasible to obtain these data, which could strengthen the conclusion that these two proteins act as nuclear/nucleolar proteins in gene expression regulation.

> Figure S2C: explain the difference between the expected size (1146 bp) and detected size (700 bp) of PCR product

> Line 147-152. The transmission of the parasite from the human host to the mosquito vector is from 37°C to 22-25°C, the latter is commonly used for malaria-infected mosquitoes in the lab facility. The author cultured the asexual blood parasite at 32°C. This mimic is far from the physiological condition for the parasite transmission.

> L154-155 and Fig S3C. Different contents were described in the text and figure, it is not match.

> L156-157: The HMGB1 enrichment at rRNA locus at 32°C over 37°C was shown only for S3 locus, but not for S1 and S2 loci. Please provide the results for S1 and S2 loci.

> Figure S3I: Explain the difference between the expected size and detected size of PCR product

> Figure S2D, Figure S2E, Figure S3K, and Figure S6I. The protein marker showing the size of Aldolase is different in different blots!

> Line 208-209 and Figure S4B. The images of the Geimsa-stained parasite were cited in the text, but do not support the conclusion (de-repression of rRNA) in the text.

> Line 217, NAD⁺ is an essential cofactor for Sirtuins, not for the HDACs.

> Line 215, 290, 354. Sirtuins and HDAC are different kind of deacetylase. It is not right to use the “HDAC Sir2a”.

> In Figure S6C. Sir2a is expressed as a nuclear protein in the asexual blood stage, but the author failed to detect the Sir2a-HA in the nuclear fraction. The author generated the Sir2a-madID parasite line, but did not provide any evidence showing the nuclear localization of the Sir2a-madID protein.

> Line 277-279. In direct comparison to the active rDNA locus, however, Sir2a occupancies are 2- to 20-fold higher upstream of all major silent rDNA loci (Figure 3H, S7G). How did the author calculate the 2- to 20-fold? Have no idea from Figure 3H and S7G.

> L341. Figure S7F is a schematic of the working model, but it was cited here. Wrong?

> Many errors in the figure and text.

Thanks for the update in the revised manuscript by the authors.

> One major concern remains unaddressed.

Throughout the manuscript, the author only emphasized the upregulation of S-type (S1, S2, and S3) rRNA in the asexual blood stage in temperature or metabolic manipulations, but did not fully present the extent or level of the rRNA upregulation.

In the Supplementary Table 4, the results showed that the transcript levels for the S-type (S1, S2, and S3) rRNA are 30000-50000 (in FPKM) in oocyst and sporozoite. These data show the real physiological level for S-type (S1, S2, and S3) rRNA expression. In either the introduction or discussion section of the manuscript, this information is not included. It is necessary to highlight and describe these information in the manuscript (for the full understanding of readers).

In the Supplementary Table 4, the results also showed that after either 32°C temperature or knockdown of Nico, LDH, or Sir2a, the upregulated levels (ranging from 25 to 254) of S-type rRNA are still quite lower than the level (30000-50000 in FPKM) of S-type rRNA in oocyst and sporozoite. Even after a 10-fold upregulation, it is still less than **1%** of the physiological expression level (30000-50000 in FPKM) in the oocyst and sporozoite stages!

Figure 1 A,B; Figure 2A																
Name	ID	NF54-32C R1	NF54-32C R2	NF54-32C R3	NF54-37C R1	NF54-37C R2	NF54-37C R3	Gam R1	Gam R2	Gam R3	Ooc R1	Ooc R2	Ooc R3	Spz R1	Spz R2	Spz R3
S1	PF3D7_011270	237	255	256	70	65	64	59	48	64	31433	29732	30781	60742	56230	55878
A1/2	PF3D7_072600	120220	124913	118601	124439	126461	124376	112496	109347	112927	105039	107670	104143	65809	76260	75089
S2/3	PF3D7_114860	214	238	250	12	14	16	15	8	9	38588	35021	36251	41893	38369	36736

Figure 2F		no glucosamine			with glucosamine		
Name	ID	Nico-gln	Nico-gln	Nico-gln	Nico-gln	Nico-gln	Nico-gln
S1	PF3D7_011270	45	53	34	87	66	75
A1/2	PF3D7_072600	130717	118416	131844	126253	128013	127714
S2/3	PF3D7_137130	4	3	8	20	6	24

Figure 2H		no glucosamine			with glucosamine		
Name	ID	LDH-glm	LDH-glm	LDH-glm	LDH-glm	LDH-glm	LDH-glm
S1	PF3D7_011270	44	38	46	61	64	73
A1/2	PF3D7_072600	138308	126790	136897	127856	126034	120246
S2/3	PF3D7_137130	4	7	9	19	25	29

Figure 2I		WT-3D7			Sir2a-KO		
Name	ID	WT-3D7	WT-3D7	WT-3D7	Sir2a-KO	Sir2a-KO	Sir2a-KO
S1	PF3D7_011270	37	47	52	105	83	94
A1/2	PF3D7_072600	131756	126893	123845	130228	129369	126109
S2/3	PF3D7_137130	7	8	5	13	19	13

The author showed that the environmental temperature switch or metabolism manipulation in the asexual blood stage parasite has a minor effect on the upregulation expression of S-type rRNA, but this change in transcript level is far from the real physiological level in the oocyst/sporozoite stage. For this point, I was not convinced by the update in the revised manuscript.

> Some cited references from the preprint have been officially published in the journal. Need updated.